# Motion-corrected eye tracking improves gaze accuracy during visual fMRI experiments

Jiwoong Park [1,2,3], Jae Young Jeon [1,3], Royoung Kim [1,3], Kendrick N. Kay [4] & Won Mok Shim [1,2,3] ✉

Human eye movements are essential for understanding cognition, yet achieving high-precision eye tracking in functional Magnetic Resonance Imaging (fMRI) remains challenging. Even slight head shifts from the initial calibration position can introduce drift in eye tracking data, leading to substantial gaze inaccuracies. To address this, we present Motion-Corrected Eye Tracking (MoCET), which corrects drift using head motion parameters derived from fMRI preprocessing. MoCET requires no additional hardware and can be applied retrospectively to existing datasets. We show that it outperforms conventional detrending methods with respect to accuracy of gaze estimation and offers higher spatial and temporal precision compared to magnetic resonance-based eye tracking approaches. By overcoming a key limitation in integrating eye tracking with fMRI, MoCET enables precise investigations of naturalistic vision and cognition in fMRI research.

Human eye movements are closely linked to cognitive processes such as perception, attention, and memory[1–3]. Acquiring eye tracking data during functional magnetic resonance imaging (fMRI) experiments offers the possibility of better understanding these processes at both neural and behavioral levels. By combining eye tracking with fMRI data, we can link explicit behavioral signatures of mental states to underlying neural activity, thereby providing insights into how the brain responds to the visual world[4–8]. This integration becomes increasingly crucial as naturalistic free-viewing paradigms gain prominence in cognitive neuroscience: in naturalistic environments, precise eye tracking is necessary to understand the brain's dynamic processing of visual information[9–12].

Recent advances in eye tracking methods within fMRI research typically involve camera-based systems using endoscopic optic fiber[13,14]. These systems usually monitor one eye with a single camera, using computer vision techniques to detect pupil position. While this approach enables precise eye tracking data, it faces major technical challenges, such as hardware setup constraints in the MRI environment and limitations in calibration data quality.

Camera-based eye tracking accuracy relies critically on calibration that maps pupil positions to gaze positions. The MRI scanner's limited visual field makes precise calibration difficult, as participants fixate closely-spaced calibration points. More importantly, even slight head movements from the calibration position compromise accuracy. Unlike behavioral experiments that can use chin rests or bite bars, such head restraints are not feasible in MRI. Consequently, head shifts significantly reduce gaze accuracy, as the calibrated model is misaligned with eye tracking data collected during the experiment[15–17].

Behavioral experiments have recently introduced several methods for addressing head shifts, such as recording head movements through supplementary cameras, motion sensors, or a wide-field camera that captures both the participant's eye and head, allowing for separate tracking of eye and head movements[18–20]. However, these solutions are challenging to implement in fMRI experiments. Installing additional cameras or motion sensors in the head coil is not feasible in most cases, and the narrow field of view (FOV) of the camera, focused on a single eye, limits the ability to capture head movements. This challenge emphasizes the need for new solutions that maintain

[1]Center for Neuroscience Imaging Research, Institute for Basic Science, Suwon, Republic of Korea. [2]Department of Biomedical Engineering, Sungkyunkwan University, Suwon, Republic of Korea. [3]Department of Intelligent Precision Healthcare Convergence, Sungkyunkwan University, Suwon, Republic of Korea. [4]Department of Radiology, Center for Magnetic Resonance Research, University of Minnesota, Minneapolis, MN, USA. ✉e-mail: wonmokshim@skku.edu

accuracy despite the inevitable head movements that occur in fMRI experiments.

In this study, we investigate the impact of head motions on gaze accuracy during fMRI scans. We first develop computational simulations that confirm head movements significantly affect gaze estimation accuracy. Based on this finding, we then propose an eye tracking with head motion correction approach, which addresses head movement challenges without requiring additional hardware and can be retrospectively applied to existing data. Our approach, termed Motion-Corrected Eye tracking (MoCET), leverages head motion parameters derived from the preprocessing of fMRI data to correct errors in eye tracking data.

Using high-quality eye tracking data collected during free-viewing tasks, we demonstrate that MoCET consistently outperforms both uncorrected data and conventional detrending methods across independent datasets involving different tasks and MRI scanners. We further validate MoCET by showing that its corrected gaze accuracy closely matches that of behavioral eye tracking data collected under standard laboratory conditions. To enhance flexibility, we introduce an across-run variant of MoCET that enables calibration transfer across runs within the same session, allowing robust correction even when within-run calibration is limited or unavailable. Finally, we compare MoCET with magnetic resonance-based eye tracking[21], which estimates gaze position directly from eyeball region signals in fMRI data. We find that while MR-based methods successfully capture general gaze direction, they lack spatial precision. In contrast, we show that MoCET provides precise gaze locations, making it more suitable for detailed behavioral and neuroimaging analyzes.

To facilitate the application of MoCET across diverse fMRI experiments, we provide a user-friendly Python package (https://github.com/jwparks/mocet). Additionally, we release the high-quality free-viewing eye tracking dataset used in this paper. This dataset, containing multiple calibration periods, enables both calibration and validation of eye tracking models, and may serve as a useful benchmark for advancing eye tracking methods, including MR-based eye tracking[21–24].

## Results

### Impact of head motion on gaze tracking accuracy

We simultaneously collected fMRI and eye tracking data while participants engaged in interactive Minecraft[25]-based video game tasks (Fig. 1a). These tasks allowed free gaze movement (instead of central fixation) and were designed to provide a diverse range of visual stimuli and cognitive demands, enabling assessment of gaze tracking accuracy under different conditions. To assist in eye tracking, two eye tracking calibration stages were included during the actual data collection. For each stage, participants fixated on sequentially appearing green dots. The first calibration stage involved 24 dots (two repetitions of a 12-dot sequence), while the second validation stage occurring at the end of the experiment required fixation on 12 dots. The data collected from the initial calibration stage were used to fit our eye tracking model (Fig. 1b).

We hypothesized that small head movements during fMRI sessions prevent the calibration model from accurately tracking pupil positions. Specifically, in-plane head movements (those occurring in the $p_x$-$p_y$ plane perpendicular to the eye tracking camera) shift the recorded pupil coordinates, creating discrepancies between the calibration and validation stages. These shifts lead to inaccuracies in mapping pupil positions to their corresponding screen locations. Importantly, we found that even minor head shifts typical in fMRI experiments can cause substantial inaccuracies in eye tracking results (Fig. 1f, $r = 0.969$, $p < 0.001$).

Head motion is a recognized challenge in gaze tracking[15–17], often addressed by measuring head motion independently from eye movement, using motion sensors or video processing of visual features.

However, implementing additional head motion sensors within an MRI scanner is technically challenging. Moreover, the limited FOV of endoscopic eye tracking cameras prevents capturing stable feature points, such as the nose, needed for head position estimation. Features visible within the camera's FOV, like the lacrimal caruncle or eyelid, are highly sensitive to blinks and eye movements, rendering them unreliable for head motion estimation. These limitations highlight a critical need for an effective method to capture and compensate for head motion, ensuring accurate gaze tracking throughout fMRI experiments.

To rigorously demonstrate that participant head motion is the primary source of eye tracking errors rather than errors in instrumental setup or suboptimal fixations during calibration, we developed a computational simulation using a 3D geometry-based eyeball model. This approach offers a controlled framework to isolate and quantify the impact of head motion on gaze accuracy. The model included head and eyeball components that could each move independently. The head-ball component allowed six degrees of freedom (DoF): translation along $x$, $y$, and $z$ axes, and rotation including roll, pitch, and yaw. The eyeball was physically connected to and moved with the head but was able to rotate independently to simulate gaze direction. Our simulation recorded pupil position on the eyeball using a virtual eye tracker and applied the same analysis procedures used for human participants (e.g., extract the pupil position, fit the eye tracking calibration model).

To simulate the impact of human head motion on gaze accuracy, we applied 6-DoF head motion parameters sampled from actual fMRI data to our model. The model performed an initial calibration task (same as human subjects) and then performed a central fixation task for about 13 minutes. (During this central fixation task, changes in pupil coordinates directly reflected the applied head motion effects. Next, the model performed a validation task, which was used to evaluate gaze accuracy (Fig. 2a). The model parameters (head/eyeball radius, eye location) matched average Asian head proportions[26–28], while the FOV of the camera, screen size, and eye-to-screen distance closely replicated our human fMRI experiment setup (See Supplementary Fig. 1 and Methods).

Our eyeball simulation provides two key insights: First, comparing simulated and human gaze trajectories reveals the amount of variance in human eye tracking data that is attributable to head motion. Similar trajectories between the simulated model and human participants indicate that the observed drift in pupil coordinates is driven by head motion rather than actual eye movements. Second, although head motion affects pupil coordinates in a complex, non-linear manner, the simulation allows us to assess whether these effects can be effectively corrected using linear techniques. If effects are approximately linear, this would justify the use of fMRI-derived head motion parameters in a linear regression procedure to correct for head motion effects.

Our model simulation demonstrated that both horizontal ($p_x$, average model similarity: $r = 0.478$) and vertical ($p_y$, average model similarity: $r = 0.793$) eye movements can be replicated using motion parameters derived from participants' actual head motion, with vertical shifts ($p_y$) particularly well-aligned with model simulation due to characteristic pitching movements (Fig. 2b). To assess statistical significance, we generated randomized head motion parameters by randomly shuffling the phase spectra, and simulated eye movements based on these randomized parameters, creating a null distribution of similarity values ($n = 100$ per simulation). Comparing actual simulation similarity to this distribution, with results aggregated across datasets using Fisher's method[29], we find statistical significance for both horizontal and vertical pupil coordinates ($p < 0.001$), indicating that head motion drives global error in human eye movements.

We found that the model simulation accurately replicated both the magnitude of head shift from initial calibration to final validation and the resulting gaze inaccuracies caused by head motion in

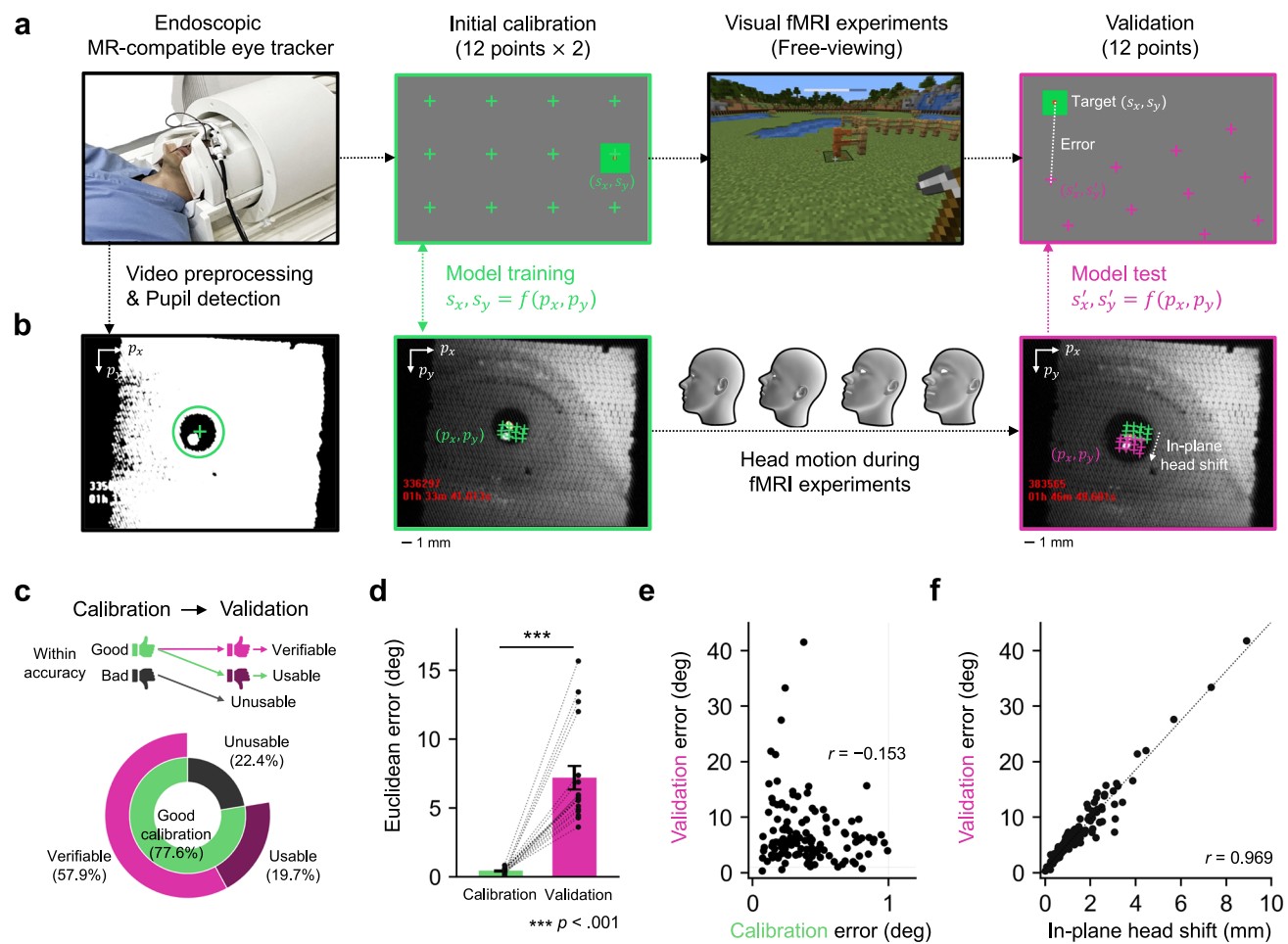

**Fig. 1 | Eye tracking in fMRI experiments. a** An endoscopic eye tracking camera captures participants' eye movements throughout the experiment. At an initial calibration stage at the beginning of each experiment, 12 calibration targets are presented sequentially on the screen, and participants are instructed to fixate on these targets. Gaze accuracy is evaluated at the end of the experiment in a validation stage by measuring the distance between the predicted gaze location and the actual target location. **b** Eye tracking video is preprocessed using low-pass filtering and binarization to isolate the pupil and remove spatial noise. Pupil coordinates recorded during the calibration stage are used to train the eye tracking model. In-plane head shifts, occurring within the $p_x$-$p_y$ plane, are estimated by comparing pupil coordinates across the calibration and validation stages. **c** Proportions of runs classified as *verifiable*, *usable*, or *unusable* based on calibration accuracy. Verifiable runs were used for all analyses, usable runs for analyses that rely on calibration, and unusable runs only for across-run generalization. See Methods for criteria and usage in analyses. **d** Gaze accuracy significantly decreases during the validation stage ($t_{paired}$ (17) = 7.95, $p < 0.001$). Error bars indicate ± 1 s.e.m. across participants ($N = 18$). **e** No significant positive relationship is observed between calibration error and validation error. **f** The extent of in-plane head shifts shows a strong linear relationship with eye tracking error during the validation stage. **e**, **f** Each arrow and dot represent eye tracking data from individual runs across all participants. Source data are provided as a Source Data file. The video game screenshot is used under the Minecraft Usage Guidelines (©2025 Mojang AB; Minecraft® is a trademark of Microsoft Corporation).

individual human eye tracking data. Head shifts in the simulation showed a strong linear relationship with estimated human head shifts ($r = 0.917, p < 0.001$). Similarly, the simulation reliably reproduced individual gaze inaccuracies, showing a strong positive correlation between simulated and actual gaze errors ($r = 0.958, p < 0.001$). While human gaze errors were slightly larger than simulation gaze errors ($t_{paired}(132) = 12.75, p < 0.001$) this likely reflects the fact that actual human fixations have limited accuracy.

These findings highlight the effectiveness of the geometry-based model in isolating head motion effects on eye tracking data. By factoring out task-related variability in eye position and human-specific idiosyncrasies, the simulation confirms that even minor head movements can introduce substantial inaccuracies in eye tracking data.

## Compensating for head movement with motion-corrected eye tracking

Our model simulation demonstrated that head motion alone can cause global drift in eye tracking data, even in the absence of actual eye movement. While many MR-compatible eye trackers offer drift correction, these methods assume that participants fixate on known locations during the experiment. This assumption fails in free-viewing experiments such as movie-viewing or playing video games, since the participant's gaze is not restricted to predefined targets. Without fixed fixation points, conventional drift correction cannot determine the offset between recorded and actual gaze positions.

Based on our computational simulation, which showed that head motion introduces systematic, approximately linear error in pupil coordinates, we developed a correction approach. Our method, MoCET, uses head motion parameters as nuisance regressors to account for motion-induced drift, thereby helping to isolate actual eye movements (Fig. 3a).

MoCET uses 6-DoF head motion parameters from fMRI data and polynomial regressors (up to the cubic order) to remove error in eye tracking data. The polynomial regressors address very low-frequency drift not fully captured by head motion parameters, such as instrumental shifts and vibrations in the MRI environment affecting the

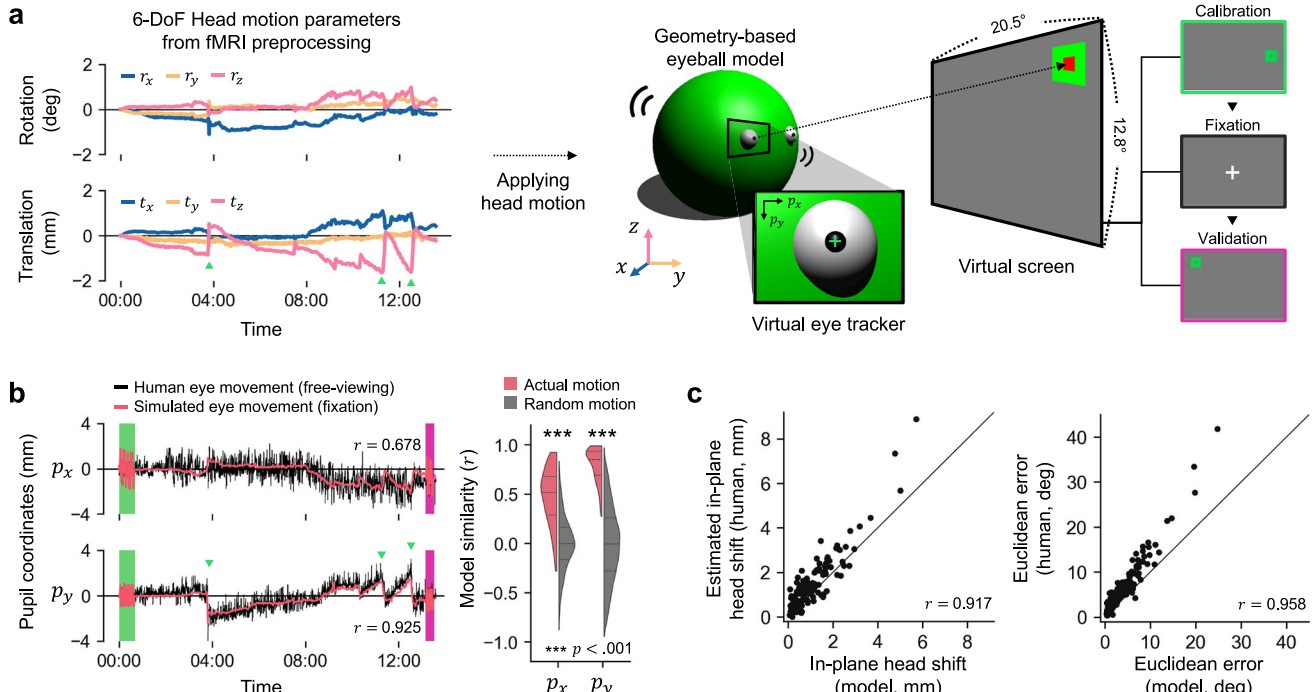

**Fig. 2 | Computational simulation of eye tracking during the influence of head motion. a** Six-degree-of-freedom (6 DoF) head motion parameters derived from the preprocessing of fMRI data were used to simulate eye movements with a geometry-based eyeball model. The model performed a calibration task requiring fixation on 12 calibration points on the screen (repeated twice), a central fixation task, and a validation task requiring fixation on the same 12 points. Simulated eye movements were rendered as images, and pupil coordinates were tracked using the same method applied to human participants. **b** Comparison of human and simulated eye movements. Despite the model performing a central fixation task, its global eye movement trends closely resembled those observed in human eye tracking data. Abrupt changes in the vertical pupil coordinate ($p_y$), indicated by green triangles, occurred in opposite directions to vertical head motion along the $z$-axis as shown by translational motion in (**a**). The initial calibration and validation stages are highlighted by green and magenta rectangles, respectively (Left).

Simulated horizontal and vertical pupil coordinates derived from participants' actual head motion demonstrated significantly higher similarity to human data compared to simulations using randomized head motion generated by phase-shifting permutation (Right). Statistical significance was determined by comparing actual values to the null distribution of similarity with a one-sided $p$-value, with horizontal lines indicating quantiles. **c** The model's estimated in-plane head shift (perpendicular to the eye tracking camera) exhibited a strong linear relationship with the estimated in-plane head shift observed in human data (Left). Simulated gaze inaccuracies were also positively correlated with those in human data, although human errors were generally larger due to the imprecision of actual human fixations (Right). Each dot represents eye tracking data from individual runs across all participants, and the diagonal line ($y = x$) represents perfect correspondence. Source data are provided as a Source Data file.

stability of the eye tracking camera. These drifts, unrelated to head motion, cannot be corrected by head motion parameters alone. By using both motion parameters and polynomial regressors, MoCET provides comprehensive correction. Head motion parameters effectively handle dynamic changes associated with head movements (high-frequency), while polynomial regressors address low-frequency drift from other sources unrelated to direct head motion, ensuring both dynamic and gradual error are corrected.

To assess the validity of using linear regression to remove head motion-related error, we compared the regression weights of six head motion parameters between human data and simulation containing only motion-related error. In both human and simulated data, horizontal head motions, such as translational movement along the $x$-axis (left-to-right) and rotational movement around the $z$-axis (rolling), primarily affected global drift in the $x$-axis of pupil coordinates, while vertical head motions, including translational movement along the $y$-axis (up-and-down) and rotational movement around the $x$-axis (pitching), affected vertical drift in the $y$-axis (Fig. 3b). These results indicate that the overall trends in regression weights across the six head motion parameters were consistent between human and simulation data. However, $y$-axis rotation (yawing) showed stronger effects in $x$-direction drift in human eye movement than in simulation, likely because the human brain sits above the eyes, unlike our model, where the yawing center of the head was close to the eye position, allowing yawing motion to have a greater influence on horizontal eye position.

To further test the linear assumption, we evaluated the non-linearity in the relationship between head motion and pupil coordinates across different types of head motion. We found that the linear and nonlinear variants of MoCET performed comparably across motion categories. Additional simulations with artificially amplified motion indicated that the linear MoCET model remained effective at motion levels typical of fMRI studies, while nonlinear models offered only limited additional benefit under extreme motion (for details, see Supplementary Fig. 3 and Methods). Overall, the regression-based approach used in MoCET successfully removed effects of physical head motion in both human and simulated eye tracking data.

## MoCET enables high-precision eye tracking during free-viewing experiments

Global drift in eye tracking data during fMRI experiments is a well-known issue and is commonly addressed using heuristic detrending methods such as linear[30,31] or quadratic polynomial detrending[21]. These methods assume that the average gaze position remains centered on the screen and drift occurs gradually rather than abruptly. A limitation of these methods is that they cannot correct transient effects such as those driven by abrupt head movements. Such head movements are especially pronounced in naturalistic tasks (e.g., free-viewing and video-game playing).

We compared the effectiveness of MoCET to conventional linear and polynomial detrending methods, as well as uncorrected data. Polynomial detrending up to the cubic order served as a baseline for

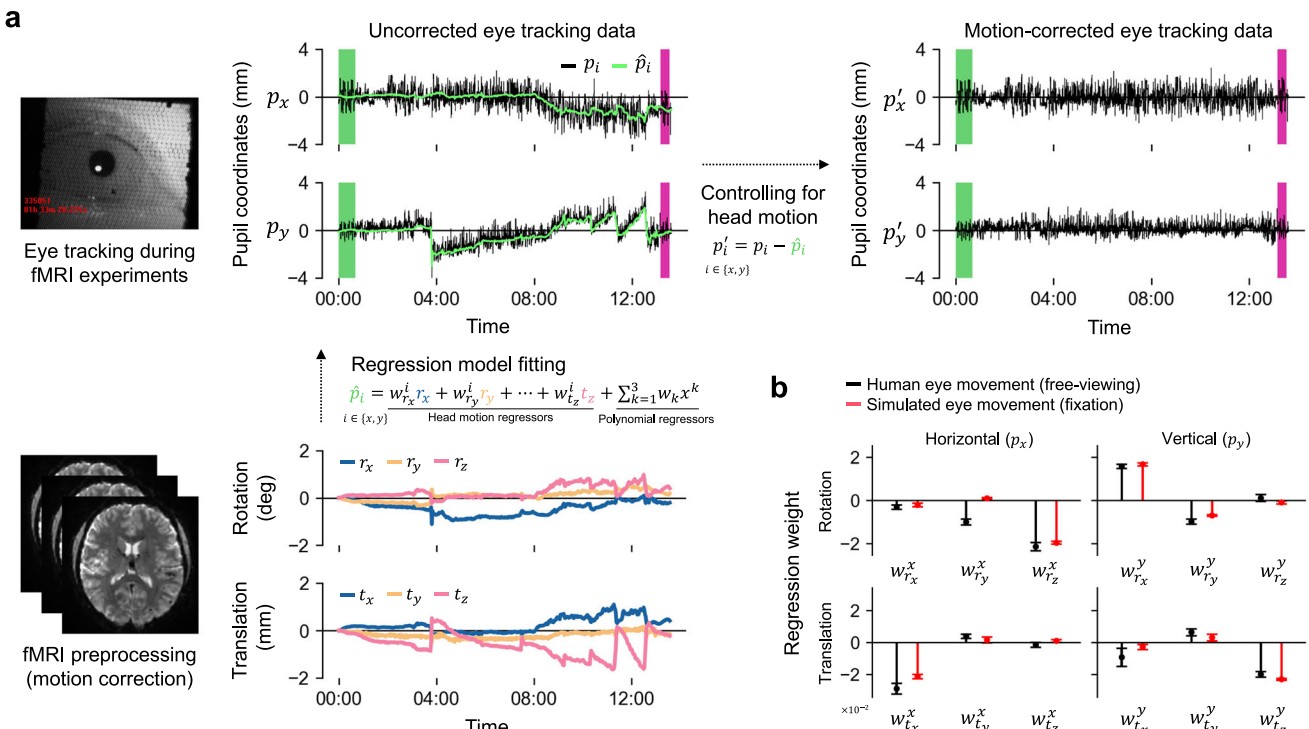

**Fig. 3 | Motion-corrected eye tracking (MoCET) pipeline. a** MoCET uses head motion parameters to compensate for the effects of head motion on eye tracking data. A combination of head motion parameters and polynomial regressors is used to model the global trend of pupil coordinates ($\hat{p}_i$). After subtracting the fitted global trend, the motion-corrected eye tracking data exhibit a more stable structure while retaining variance (jitter) from actual eye movements. **b** Comparison of regression model weights between human and simulated eye movements. The relative magnitude of regression weights reflects the strength of each head motion parameter's influence on horizontal or vertical pupil coordinates. For instance, rotational motion along the $x$-axis (i.e., pitching motion) has a strong positive effect on vertical pupil coordinates ($w^y_{r_x}$) in both human and simulated eye movements. Error bars represent ±1 s.e.m. across participants ($N = 18$). Source data are provided as a Source Data file.

comparison, while MoCET included additional 6-DoF head motion parameters. To evaluate drift correction performance, we employed three distinct metrics. First, we measured gaze accuracy during the validation stage, which evaluates eye tracking data by requiring participants to fixate on specific points across the screen (Fig. 4a). However, this metric cannot fully assess the quality of eye movements during the actual experiment and requires an additional validation stage. Thus, we introduced two additional performance metrics that do not require a validation stage and are also suitable for free-viewing experiments: the accuracy of eye tracking data in predicting participant behavior during the task (Fig. 4b) and the accuracy of mapping visual stimuli to retinotopic representations in the visual cortex (Fig. 4c, d). The latter constitutes a stringent test of eye tracking accuracy, since retinotopic representations are expressed relative to the exact gaze location.

For both model simulation and human eye movement data, MoCET outperformed other methods in reducing gaze inaccuracy during the validation stage (Fig. 4a, all $p$s < 0.001). To ensure the observed improvements were not simply due to MoCET's use of a total of nine regressors compared to polynomial detrending's three, we included additional polynomial detrending regressors up to the 12th order. Results showed that, for both model simulation and human eye tracking data, the performance of polynomial detrending saturated after the 5th order. MoCET demonstrated superior effectiveness in drift correction, even when using fewer regressors compared to 12th-order polynomial detrending (Supplementary Fig. 2, Model simulation: all $p$s < 0.01, Human eye movement, all $p$s < 0.001). Notably, gaze accuracy achieved by MoCET was not statistically different from that observed in behavioral eye tracking experiments conducted outside the scanner, indicating that MoCET recovers high-quality gaze signals even under the constraints of fMRI environments ($t(20) = 1.34, p = 0.195$).

To evaluate drift correction performance throughout the entire fMRI experiment, we assessed the ability of gaze data to predict participant behavior during the task. In the Minecraft-based video game task, participants were required to build or remove fence blocks inside a square arena. Previous research has shown that humans tend to fixate on the next target location before performing an action toward it[1,32–35]. Therefore, we hypothesized that motion-corrected gaze data should be able to predict participants' next building action or target-removal action before the action occurs. We calculated the average hit rate over the 10-s period leading up to each action. A "hit" was defined as the participant's gaze falling within 2 degrees of visual angle from the next target location. While both linearly detrended and uncorrected gaze data performed at near chance levels, MoCET demonstrated significantly better prediction performance compared to all other detrending methods (Fig. 2b; all $p$s < 0.001). We also demonstrated that the hit rate from MoCET was comparable to that observed in behavioral eye tracking, indicating reliable recovery of task-relevant gaze patterns during fMRI experiments ($t(20) = 1.40, p = 0.175$). These results suggest that the error correction achieved by MoCET effectively removes non-gaze-related artifacts, such as head motion or instrumental instability, while preserving actual eye movements throughout the task.

As an additional performance metric, we evaluated the ability of gaze data to enable the derivation of retinotopy during free-viewing conditions. Retinotopic mapping in the visual cortex refers to the spatial organization of neural representations of the visual field. Each region of the visual field is systematically represented in early visual areas, such as V1, V2, and V3, with maps that are defined relative to the gaze location[36,37]. This retinotopic organization provides a critical link between eye movements and neural activity, and deriving retinotopic organization during free-viewing requires high-accuracy gaze data. To

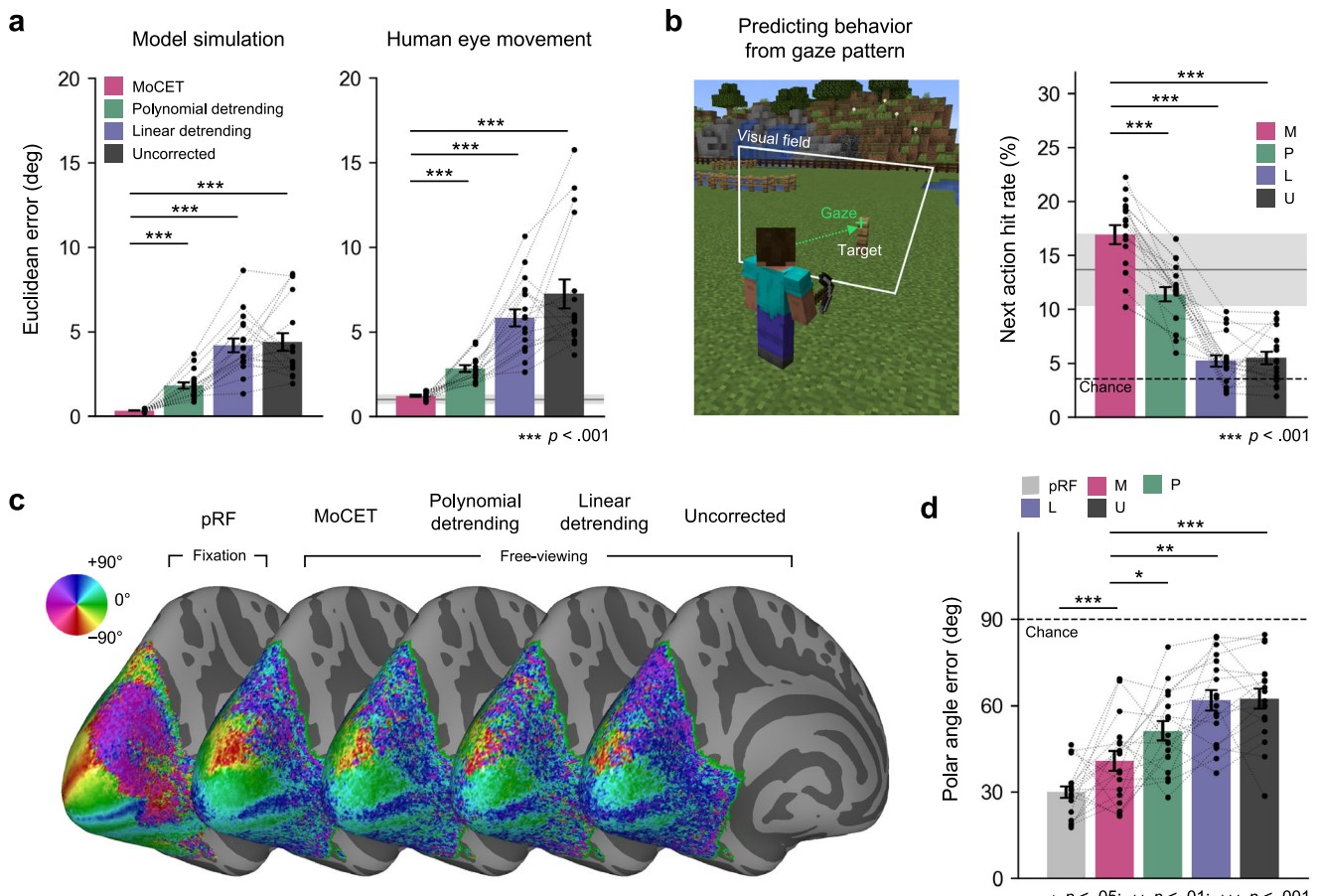

**Fig. 4 | Comparison of drift correction performance between MoCET and conventional heuristic detrending methods. a** MoCET significantly outperformed conventional detrending methods in reducing eye tracking error during the validation stage, as demonstrated in both model simulation data (Left) and human eye tracking data (Right). Consistent results were observed in two independent replication datasets (Supplementary Fig. 4). **b** MoCET showed superior performance in predicting subsequent participant behavior based on gaze patterns during the task. Specifically, while participants sequentially built or removed blocks in a Minecraft environment, the hit rate was measured as the percentage of time participants gazed at the next target location before performing an action at the target. Chance level was calculated as the percentage of random gaze locations on the screen coinciding with the next target location. **c** Group-averaged polar angle representations in visual cortex. Using population receptive field (pRF) modeling, polar angle representations were estimated based on data from the pRF experiment with retinotopic stimuli and data from the free-viewing experiment. While the conventional pRF experiment, in which visual stimuli stimulate specific retinotopic locations during central fixation, provided the clearest polar angle maps, MoCET exhibited the best performance in deriving retinotopic maps from free-viewing data. **d** Polar angle accuracy was assessed by calculating the error between predicted polar angles from pRF modeling and canonical polar angles. Ten cortical landmarks with representative polar angles (e.g., calcarine sulcus and V1-V2 boundaries) were used for this analysis. MoCET applied to free-viewing experiments showed significantly better performance in mapping retinotopic visual fields compared to other methods. Chance level was determined as the average error from randomly selected angles (0–360°) compared to the target angle. Error bars indicate ±1 s.e.m. across participants. **a, b, d** Each dot represents one participant, and gray shaded areas in (**a, b**) indicate ±1 s.e.m. across four participants from independent behavioral eye tracking experiments. All performance metrics were compared across methods using paired t-tests with two-sided p-values. Source data are provided as a Source Data file. The video game screenshot is used under the Minecraft Usage Guidelines (©2025 Mojang AB; Minecraft® is a trademark of Microsoft Corporation).

transform free-viewing stimuli into gaze-centered stimuli, we corrected eye tracking data using various drift correction methods (MoCET, polynomial, linear detrending, and uncorrected data), and then calculated the spatiotemporal local contrast of the stimulus within small patches defined relative to gaze location[38,39]. Using this gaze-centered stimulus preparation, we fit a population receptive field (pRF) model[40,41] to recover spatial tuning (i.e., polar angle and eccentricity) in the early visual cortex (for details, see Supplementary Fig. 5 and Methods). Results from free-viewing stimuli were compared to spatial tuning obtained from conventional pRF experiments using structured visual stimuli (wedges, rings, and bars) presented during central fixation.

While pRF experiments with structured stimuli provided the clearest polar angle maps, MoCET demonstrated the ability to accurately map free-viewing stimuli to brain activity and outperformed other detrending methods in recovering polar angles (all ps < 0.05). Although the video game stimuli showed limitations in mapping foveal

representations, they more effectively mapped peripheral visual areas compared to the structured pRF experiment, even along the vertical meridian, where stimulus coverage was comparable to pRF stimuli. This advantage stems from free-viewing experiments naturally engaging the peripheral visual field, whereas structured pRF experiments conducted under central fixation are typically limited in their stimulation extent. Detailed retinotopic representations, including eccentricity maps, are presented in Supplementary Fig. 6.

In summary, our results demonstrate that MoCET significantly improves eye tracking accuracy during free-viewing experiments compared to conventional detrending methods. By combining head motion parameters and polynomial regressors, MoCET not only reduces drift-related errors but also enhances the predictive power of gaze data for behavior and neural response characterization. MoCET consistently outperformed other methods in gaze accuracy during validation stages, behavioral prediction during dynamic tasks, and retinotopic mapping of

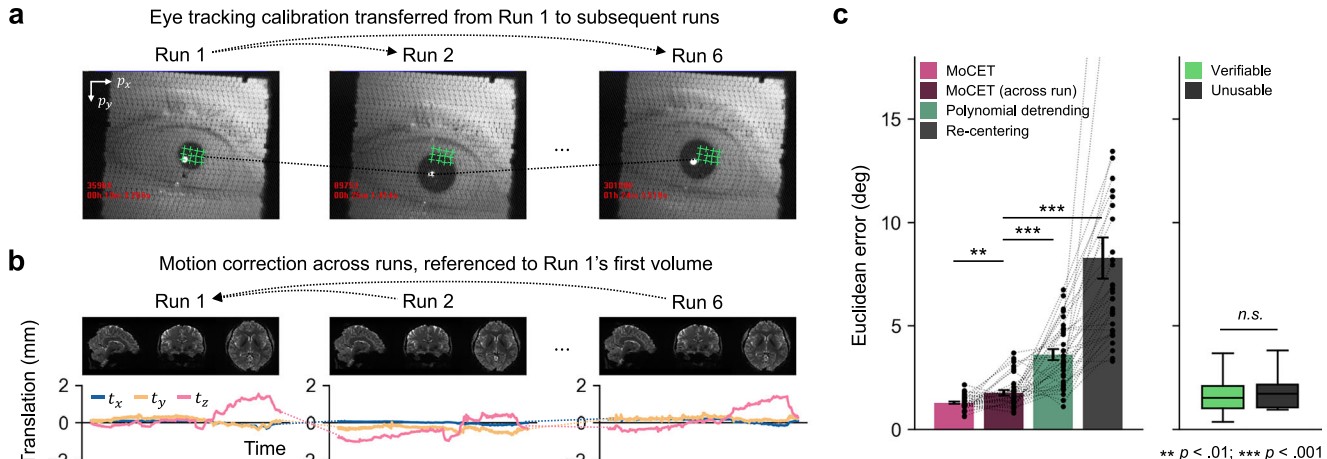

**Fig. 5 | Robust correction of across-run drift using MoCET. a** Pupil center locations across six consecutive runs from a representative session. Large inter-run shifts highlight the limitation of applying a Run 1's calibration denoted as green crosses to subsequent runs without accounting for head motion. **b** Head motion parameters from each run, referenced to the first volume of Run 1. Only translational components are shown here for illustrative purposes. **c** Comparison of gaze accuracy across methods. Across-run MoCET performs slightly below within-run MoCET but substantially outperforms baseline methods. Each dot represents a scanning session from all participants. Error bars represent ±1 s.e.m. across participants (Left, N = 34). Across-run MoCET performance did not differ between sessions with usable calibration (e.g., verifiable runs, N = 106) and those with unusable calibration (Right, N = 11). Boxplots represent the distribution across runs, showing the minimum, first quantile, median, third quantile, and maximum. Source data are provided as a Source Data file.

free-viewing visual stimuli, particularly in peripheral visual areas. These findings establish MoCET as a robust and versatile method for correcting eye tracking data, enabling more precise studies of behavioral and neural processes in naturalistic contexts.

## MoCET generalizes across runs without repeated calibration
Thus far, our results demonstrated that MoCET substantially improves eye tracking accuracy within individual runs by correcting for intra-run head motion. However, head motion can also occur during inter-run intervals, such as brief resting periods or participant repositioning between scanning blocks. If eye tracking calibration is performed only once at the beginning of the experimental session, any shift in head position across runs may result in systematic errors that persist throughout subsequent runs (Fig. 5a). Conventional detrending methods or single-point re-centering strategies may not sufficiently address these across-run misalignments, particularly when rotation or non-translational motion is involved.

To evaluate whether MoCET can also correct across-run motion drift, we trained a motion-to-gaze regression model on a single calibration run (the run with the highest-quality calibration among the six) and applied it to all other runs using head motion parameters realigned to the first volume of the calibration run (Fig. 5b). As comparison baselines, we tested two alternatives: (1) a re-centering approach that re-aligns the calibration origin to the pupil center at the start of each new run, mimicking the manual drift correction used in eye tracking software; and (2) Re-centering and polynomial detrending of the raw gaze time series.

Our results show that across-run MoCET slightly underperforms within-run MoCET in overall accuracy, yet significantly outperforms both detrending and re-centering methods (Fig. 5c, all ps < 0.001). While re-centering can partially address translational drift, it fails to account for within-run motion and rotational shifts, limiting its effectiveness. Notably, across-run MoCET achieved robust performance even when runs were categorized as unusable, suggesting that effective across-run correction is possible despite suboptimal or missing within-run eye tracking calibration (Fig. 5c). These findings highlight MoCET's potential to reduce experimental burden by requiring only a single calibration run per session, while preserving reliable gaze estimation across multiple runs.

## Comparison of camera-based and MR-based eye tracking
Magnetic resonance (MR)-based eye tracking has been explored for decades as an alternative to traditional camera-based systems[17,22–24]. This approach eliminates the need for additional eye tracking hardware, as it relies solely on the fMRI data to infer gaze. Recent advancements, such as DeepMReye[21], have advanced MR-based eye tracking by leveraging deep neural networks to decode gaze coordinates directly from MR signals with promising levels of accuracy. MR-based eye tracking not only simplifies experimental setups but also enables retrospective analysis of preexisting datasets, broadening its application across studies[21,31]. However, because MR-based methods are constrained by the spatial and temporal resolution of fMRI, it remains unclear whether they can substitute camera-based systems in applications requiring precise, moment-to-moment gaze tracking.

We compared the performance of camera-based and MR-based eye tracking methods. For the MR-based approach, we examined three DeepMReye models using (1) the original pretrained model weights (*Pretrained*), (2) pretrained weights finetuned with our dataset (*Finetuned*), and (3) weights trained from scratch exclusively on our dataset (*Scratch*). Both the Finetuned and Scratch models were trained on 24 calibration points during the calibration stage (Fig. 6a). While the *Scratch* and *Finetuned* models demonstrated low training losses and effectively captured gaze coordinates during calibration, the Pretrained model struggled to fit the training data, showing near-zero responses (Fig. 6b, Euclidean error: *Scratch*: 0.518 deg; *Finetuned*: 0.886 deg; *Pretrained*: 6.193 deg).

Although the re-trained MR-based models performed well on the calibration data, they exhibited limitations when applied to the rest of the experiments. Specifically, while the DeepMReye models successfully recovered general gaze directions (e.g., looking up, down, left, or right), demonstrating their ability to capture trends in gaze movement derived from fMRI data (Fig. 6c, d; Pearson's correlation of horizontal direction: *Scratch*: r = 0.574, *Finetuned*: r = 0.562, *Pretrained*: r = 0.443; vertical direction: *Scratch*: r = 0.623, *Finetuned*: r = 0.421, *Pretrained*: r = −0.003), they showed reduced precision in mapping gaze locations, often predicting positions that deviated significantly from the ground truth (Fig. 6c, e; Euclidean error: *Scratch*: 5.521 deg, *Finetuned*: 5.525 deg, *Pretrained*: 6.184 deg).

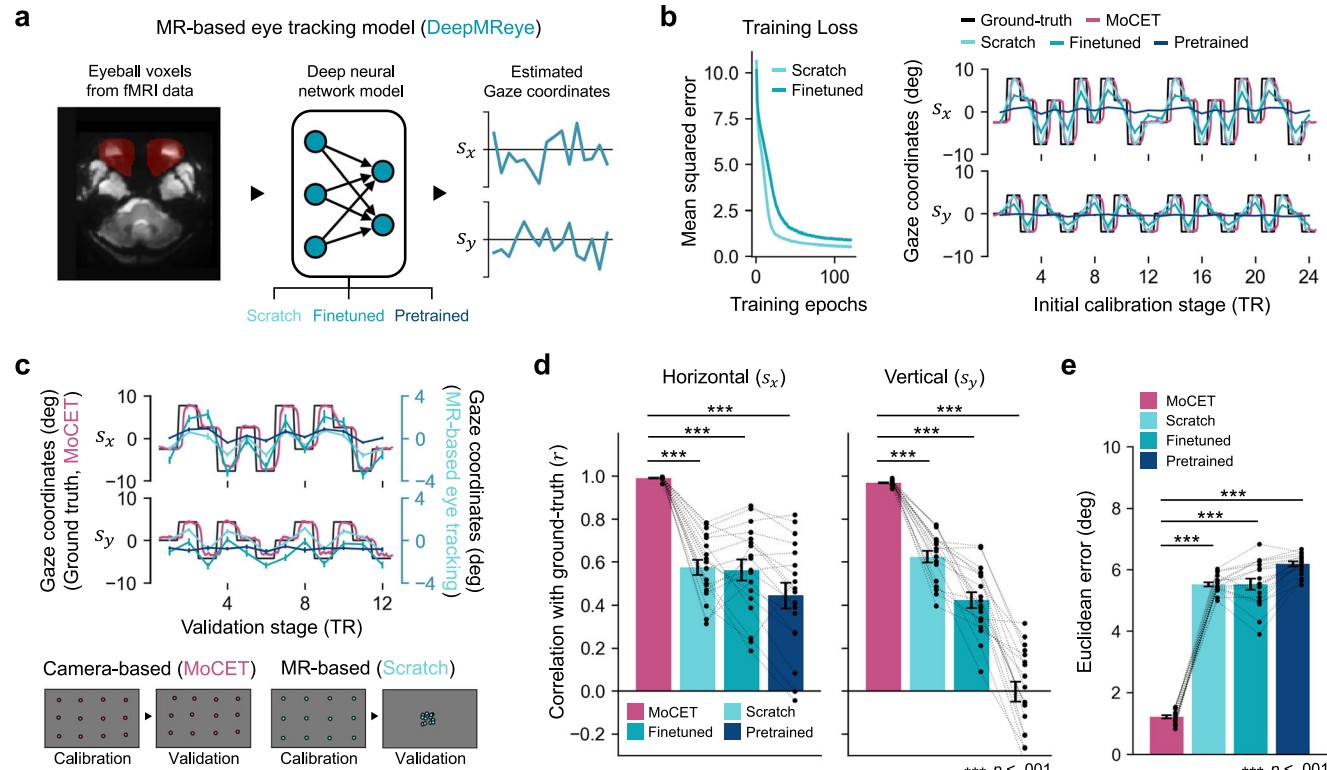

**Fig. 6 | Comparison of camera-based and MR-based eye tracking methods. a** DeepMReye[21] leverages deep neural networks to estimate gaze coordinates. Three variations of model weights were applied for subsequent analyses: (1) the original pre-trained DeepMReye model weights (*Pretrained*), (2) finetuned pre-trained weights using our dataset (*Finetuned*), and (3) model weights trained from scratch exclusively on our dataset (*Scratch*). **b** Both the Scratch and Finetuned models achieved low training losses (Left) and successfully recovered gaze coordinates during the calibration stage used for model training (Right). In contrast, the Pretrained model failed to recover accurate gaze coordinates. Notably, the camera-based eye tracking data enhanced with MoCET exhibited a clear delay relative to the onset of the calibration dots (ground truth) due to the reaction time of human participants. **c** During the validation stage, all three MR-based eye tracking models followed the general trend of the ground truth gaze coordinates, but their magnitudes were substantially reduced. While camera-based eye tracking (MoCET)

maintained spatially distributed gaze patterns close to the actual location of the dots, MR-based models produced gaze location estimates that clustered toward the center of the screen. Note that for visibility, different axis scales are used for the MR-based models. **d** For both horizontal and vertical gaze coordinates, all three MR-based models showed trends in recovering gaze direction during the validation stage. However, due to the limited vertical field of view of the screen, the *Pretrained* model showed markedly reduced performance in the vertical direction. **e** Despite significant trends in gaze direction recovery, all three MR-based models showed limitations in mapping precise gaze locations, resulting in higher Euclidean errors compared to camera-based eye tracking. **c–e** All performance metrics were compared across methods using paired *t*-tests with two-sided *p*-values. Error bars represent ±1 s.e.m. across participants (*N* = 18). Source data are provided as a Source Data file.

In contrast, the camera-based eye tracking method, enhanced with MoCET, consistently outperformed MR-based methods, maintaining both precise gaze direction and spatial accuracy of gaze coordinates across the entire experiment (*p*s < 0.001 for all comparisons). We observed that MR-based models tended to predict gaze positions near the screen's center, suggesting limited performance when generalizing to novel data. This pattern indicates that the deep neural network models had difficulty accurately mapping gaze locations beyond the distribution of their training data, leading to small-magnitude or near-zero responses when encountering unfamiliar fMRI data. The combination of high gaze direction accuracy but low spatial precision observed in our study is consistent with findings from previous studies utilizing MR-based eye tracking models[21,31]. These results suggest that while MR-based models hold promise for investigating gaze patterns related to cognitive processes in fMRI studies, they are currently less suitable for applications requiring eye tracking with high spatial and temporal precision.

## Discussion
This study highlights the critical impact of head motion on gaze accuracy in fMRI experiments and introduces MoCET as a practical and effective solution. Our computational simulation confirmed that head

motion systematically generates substantial drift in eye tracking data even without actual eye movements. By using six degrees of freedom (6 DoF) head motion parameters and polynomial regressors, MoCET demonstrated its ability to robustly correct motion-induced errors in eye tracking data, significantly outperforming conventional detrending methods.

MoCET proved highly effective in fMRI experiments requiring high-precision eye tracking data. By enabling reliable prediction of participants' future actions based on eye tracking data, MoCET demonstrates its potential for studying complex, visually guided behaviors in fMRI experiments. Furthermore, MoCET enabled accurate retinotopic mapping of free-viewing stimuli, excelling in peripheral visual areas where conventional pRF experiments often face limitations. These findings establish MoCET's broad applicability in visual neuroscience research, particularly for naturalistic, free-viewing paradigms.

Beyond improving within-run eye tracking accuracy, MoCET also generalizes effectively across multiple fMRI runs using a single calibration. By leveraging head motion parameters aligned to a shared reference volume, MoCET enables accurate correction of inter-run drift without the need for recalibration, reducing experimental burden and facilitating application to datasets with limited or inconsistent

calibration. Nevertheless, while MoCET mitigates the need for frequent recalibrations, we caution against relying solely on a single calibration per session. If the only calibration run fails due to participant movement or tracking instability, recovery may be difficult. Thus, collecting multiple calibration runs throughout the session remains a recommended best practice to safeguard data quality and ensure fallback options when needed.

Moreover, MoCET can be extended to enable drift correction across sessions, offering the potential for longitudinal eye tracking alignment without session-specific calibration. However, this application requires strict control of experimental conditions, including consistent scanner setup, identical camera configuration, and alignment of imaging coverage and reference volumes across sessions. Without such consistency, head motion parameters may not accurately reflect the relative displacement between the eye and camera, undermining correction accuracy.

We compared MoCET with the recently introduced MR-based eye tracking method DeepMReye[21], which leverages deep neural networks to estimate gaze coordinates. While DeepMReye demonstrated the ability to capture gaze direction, it exhibited significant limitations in spatial precision and often exhibited a center bias. In contrast, motion-corrected camera-based eye tracking (MoCET) consistently maintained precise spatial accuracy, making it particularly well-suited for free-viewing fMRI experiments and tasks demanding high-precision gaze data.

Although MR-based eye tracking offers unique advantages, such as not requiring additional hardware, camera-based systems retain critical benefits in terms of precision and resolution. One interesting question is what specific factors might have contributed to the poor performance of MR-based eye tracking in our data. A key difference lies in the timing of fixation presentation, which was used to train and evaluate model performance. In the original DeepMReye study, each fixation point remained on the screen for 4 s, allowing five fMRI volumes per fixation (TR: 800 ms). This prolonged duration helped stabilize eyeball images, facilitating more accurate gaze predictions. In contrast, our calibration presented each fixation point for only one TR, meaning that by the time a volume was acquired, the participant's eyes may have already shifted toward the next target. This likely contributed to inaccurate gaze estimates, with a bias toward the center of the screen.

Additionally, the DeepMReye authors note potential issues with Posterior-Anterior (PA) phase encoding (see https://github.com/DeepMReye/DeepMReye/wiki), where eyeball images can be compressed, making them difficult to distinguish. While ultra-high magnetic field (7T) typically increases spatial distortion in fMRI images, visual inspection confirmed the robust quality of eyeball shapes in our preprocessed functional images. Future studies should further explore specific conditions that impact the performance of MR-based eye tracking.

MoCET's head motion correction without additional hardware particularly benefits fMRI research, where space and setup constraints often prevent installing motion sensors. The ability to be applied retrospectively to existing datasets further extends MoCET's utility, providing researchers with a versatile tool to enhance visual neuroscience research. By integrating MoCET into naturalistic experimental paradigms, it becomes possible to align neural and behavioral data more effectively, advancing our understanding of ecologically valid human cognition.

One potential concern in applying MoCET is the discrepancy between the sampling rates of eye tracking data and fMRI-derived head motion estimates. While eye tracking data have high temporal resolution, head motion parameters from fMRI preprocessing are limited by the lower sampling rate of fMRI (typically 1–2 s per volume). The slower sampling of head motion ensures that rapid eye movements, such as saccades, are not mistakenly regressed out as head motion

artifacts. At the same time, the large gap in sampling rates may raise concerns about whether head motion effects on eye tracking data are fully captured and corrected. To address this concern empirically, we systematically downsampled eye tracking data from 60 Hz to lower sampling rates and tested MoCET's performance. The results demonstrate that MoCET's accuracy remains stable down to 10 Hz, and even under extreme downsampling (to 1 Hz), it outperforms conventional detrending approaches (Supplementary Fig. 7). We note that head motion in fMRI is primarily characterized by low-frequency components, with most power below 0.1 Hz due to small voluntary and involuntary non-periodic, transient movements. Respiration-related motion introduces additional fluctuations around 0.2–0.3 Hz[42]. Thus, apparent pupil position shifts from both transient shifts in head position and respiration-related periodic head motion can be effectively corrected using fMRI-derived head motion estimates. However, rare, fast head movements within a single TR, such as a quick shift-and-return, may not produce detectable displacement in the fMRI images. In such cases, fMRI-derived motion parameters may underestimate the movement, reducing correction accuracy.

MoCET assumes a linear relationship between head motion and gaze error, which may become inadequate in cases of extreme head movement or when the eye tracking camera is installed at a steep tilt relative to the gaze direction. In such conditions, nonlinear distortions in pupil coordinates can introduce errors not fully accounted for by linear regression. However, our empirical evaluation across diverse head motion types, including large, spike-like, and drift-like movements, showed that the linear MoCET model performed as well as more complex nonlinear variants. Furthermore, simulations with systematically amplified motion confirmed that linear MoCET remains effective across a broad range of motion levels. Nonlinear extensions provided improvements only under extreme head motion conditions, which are rarely encountered in empirical experiments (Supplementary Fig. 3). These findings support the validity of the linear assumption in typical fMRI settings, where head motion is often constrained by stabilization devices such as foam padding, minimizing extreme deviations that could introduce nonlinear distortions. To accommodate potential edge cases, we also provide these nonlinear MoCET variants in our Python package, allowing users to apply more flexible correction models when needed.

Furthermore, while its retrospective nature is beneficial for existing data, it limits its use in real-time closed-loop experiments requiring gaze information[43–46]. In future work, MoCET could be adapted for real-time fMRI experiments that require precise gaze information. By directly integrating head motion parameters extracted in real time from MR image reconstruction computers, MoCET could dynamically correct gaze coordinates during ongoing experiments. This would extend its application to real-time paradigms, such as neurofeedback or task-adaptive experimental designs, where immediate gaze accuracy is critical.

Finally, a notable contribution of this work is that we release a high-quality eye tracking dataset with multiple calibration periods contained within the dataset itself. Thus, we believe this dataset could serve as a useful benchmark for developing and evaluating new eye tracking methods for fMRI experiments. The high-precision camera-based eye tracking data can also provide a reliable reference for improving MR-based eye tracking approaches. Together, we hope that MoCET and the dataset will enhance the integration of eye tracking with neuroimaging, providing crucial support for cognitive neuroscience research that requires precise gaze measurements.

## Methods

### Participants

Nineteen participants (8 females, mean age 24.02 ± 2.54 years) were recruited for the study. All participants provided written informed consent, with the study protocol approved by the Institutional Review

Board of Sungkyunkwan University (2020-09-010-001) and received monetary compensation. Participants completed 12 runs of the Minecraft task across two or three fMRI sessions, depending on individual availability. Eye tracking data were evaluated based on specific validation criteria (see Methods—Eye tracking accuracy). One participant was excluded from the analysis as none of their runs met these criteria. As a result, data from 18 participants (average number of verifiable runs: 7.39 ± 3.03, see Fig. 1c) were included in the final analysis. All 18 participants also completed an additional fMRI session involving pRF experiments, which served as a reference for validating the retinotopic mapping results.

To further evaluate MoCET's generalizability and accuracy, we analyzed three independent datasets: two movie-watching fMRI datasets collected at 3T and 7T scanners from separate groups of 12 participants each, and a behavioral eye tracking dataset from four participants who completed three runs of the Minecraft task outside the MRI scanner. The behavioral dataset enabled a direct comparison between standard laboratory eye tracking and MRI-compatible eye tracking corrected using MoCET.

### Camera-based eye tracking system

Eye tracking during fMRI was conducted using an Avotec MRI-compatible endoscopic camera and illuminator, installed in front of the participant's right eye inside both the 7T and 3T scanners, capturing eye image frames at 60 Hz. For behavioral experiments outside the scanner, pupil position was recorded at 1000 Hz using the EyeLink 1000 Plus (SR Research) and subsequently downsampled to 100 Hz for analysis. Post hoc image processing was applied to reduce high-frequency spatial noise in the eye video and to binarize the image for improved pupil detection. For accurate pupil detection and tracking, we implemented the Pupil Reconstructor with Subsequent Tracking (PuReST) algorithm[47]. The resulting pupil coordinates were then preprocessed to exclude low-confidence data caused by blinks, defined as frames with a pupil confidence score below 0.75. Additionally, we removed abnormal spikes in the pupil coordinates that exceeded three standard deviations from the local mean, which typically occurred when the algorithm mistakenly detected non-pupil dark, roundish areas during eye closure. The identical pupil detection and tracking procedure was applied to simulated eye tracking data. Note that no online or offline drift correction was applied to behavioral eye tracking data used for comparison.

### Experimental procedure

For the main analysis, we used eye tracking data collected while participants performed a 13-min 3D Minecraft[25]-based video game task in a 7T MRI scanner. The stimulus was presented on a screen subtending 20.5° × 12.8°. Each participant completed two or three sessions, depending on individual availability, to complete twelve runs in which they played the video game on different maps. At the beginning and end of each run, participants underwent eye tracking calibration, in which they fixated on green dots sequentially displayed on the screen. The initial calibration involved two repetitions of a 12-dot sequence, resulting in participants fixating on a total of 24 dots. At the end of the video game, a validation stage required participants to fixate again on the same 12 dots. During the game, participant gaze was unrestricted, allowing for natural, free-viewing behavior. The video game task included a 3-min period during which participants built new blocks and removed existing blocks in a 3D environment. Eye tracking data from this period was used to predict subsequent building or breaking actions (See Methods—Predicting behaviors from gaze patterns for more details). For comparison, a behavioral version of the Minecraft task was conducted outside the scanner, with the display size matched to the MRI setup in visual angle. This study is neither approved nor endorsed by Mojang or Microsoft.

For the retinotopic mapping analysis, participants completed an additional fMRI session of pRF experiments. During the experiment, participants maintained fixation on a central dot and pressed a button whenever they detected a change in the dot's color. The visual stimuli consisted of colorful cartoon images (toonotopy) that were viewed through moving wedge, ring, and bar apertures, designed to stimulate localized regions of the visual field[48,49]. The wedge runs used rotating wedges (counterclockwise or clockwise) to stimulate specific angular regions of the visual field, while the ring runs involved expanding or contracting concentric rings to map eccentricity. During the bar runs, bars with varying orientations (e.g., horizontal, vertical, or oblique) swept across the visual field in multiple directions. The pRF session comprised six runs, with a total duration of ~38 min.

To assess generalizability across tasks and MRI scanners, we analyzed two additional movie-watching fMRI datasets acquired from separate groups of 12 participants each, on 3T and 7T scanners. Each dataset included standard eye tracking calibration (24-point calibration at the beginning and 12-point validation at the end of each run). In the 3T dataset, participants viewed ~12 min of first person-view movie clips on a screen subtending 27.0° × 15.2°. In the 7T dataset, participants watched two 10-min runs of naturalistic movies on a screen subtending 20.5° × 12.8°.

### fMRI data acquisition and preprocessing

Functional neuroimaging data were collected using a 7T Siemens MAGNETOM Terra MRI scanner equipped with a 32-channel Nova head coil, located at Sungkyunkwan University and the Institute for Basic Science, Center for Neuroscience Imaging Research. Blood oxygenation level-dependent (BOLD) contrast was measured using T2*-weighted functional images obtained with a dual-polarity GRAPPA (DPG) sequence (voxel size: 1.5 mm isotropic; TR: 1600 ms; TE: 21 ms; FOV: 210 × 210 mm; 96 slices covering the whole brain; flip angle: 50°). High-resolution anatomical images were acquired using an MP2RAGE sequence (voxel size: 0.7 mm isotropic; TR: 5888 ms; TE: 2.44 ms; FOV: 224 × 224 mm; 320 slices; flip angle 1: 4°; flip angle 2: 5°). The same protocol was also used for the 7T movie-watching dataset, while the 3 T dataset was acquired using BOLD fMRI protocol described in our previous study[50].

Preprocessing of both functional and anatomical data was conducted using fMRIprep[51]. Since the default skull-stripping for MP2RAGE data in fMRIprep was suboptimal, the skull was manually removed prior to preprocessing. Anatomical data were then processed for intensity non-uniformity correction, brain segmentation, and surface reconstruction. Functional data underwent motion correction and were aligned to the MNI152 standard space for analysis with DeepMReye. For voxel-wise pRF modeling, functional data were maintained in their native space to preserve precise spatial resolution. No additional preprocessing steps were applied.

### Eye tracking analysis

**Eye tracking model calibration.** During the calibration period, we collected pupil coordinates at 24 points (4 × 3 grid, repeating each point twice) presented on the screen. Each calibration point appeared for 1.6 s as a green square with a central red dot. The square expanded to its largest size at 0.8 s and then began to shrink, creating a clear visual anchor for participants. To consider the saccadic delay and maximize calibration precision, we focused on a 0.5-s period during the middle of each presentation (when the square was largest) and averaged the $x$ and $y$ coordinates of the pupil within this time window. This resulted in a set of 24 averaged pupil coordinates paired with known screen coordinates for training the eye tracking model.

To predict gaze location from pupil coordinates, we employed a radial basis function (RBF) interpolation method[52–54], which is widely used for mapping spatial coordinates. RBF interpolation is particularly effective for non-linear mappings when the relationship between input

and output coordinates may not follow a simple linear path, such as mapping recorded pupil coordinates from an oblique field-of-view eye tracking camera to flattened screen coordinates. RBF interpolation calculates interpolated values as a weighted average of distances between known calibration points and input pupil coordinate, providing a smooth and continuous mapping from pupil coordinates to gaze location on the screen. The calibrated RBF model was applied to the entire set of pupil coordinates collected during the experiment, converting each pupil position to a predicted gaze location on the screen.

**Eye tracking accuracy.** To assess the accuracy of the eye tracking data, we calculated the Euclidean distance between the predicted gaze location (averaged within each 0.5-s window) and the ground truth, defined as the actual location of each calibration point on the screen. Since gaze location was averaged after applying the interpolation model, calibration error could exceed zero due to the inherent variability in gaze position within each 0.5-s calibration window.

We implemented a two-step quality control procedure to distinguish different sources of eye tracking inaccuracy, such as instrumental error, fixation instability during calibration, and head motion-induced error. Specifically, we evaluated gaze accuracy within both the calibration and validation periods using separate models trained and tested on data from the same period. Runs with gaze error below 1.0° in both periods were labeled as *verifiable* and included in subsequent analyzes related to eye tracking accuracy (e.g., Fig. 4a). If only the calibration phase met this threshold, the run was labeled as *usable* and retained for analyzes that relied only on calibration (e.g., behavioral prediction and visual field mapping; Fig. 4b, c). Runs that did not meet the threshold in either phase were labeled *unusable* but still contributed to across-run generalization analyzes (e.g., Fig. 5c), where no within-run calibration was required. The proportions of verifiable, usable, and unusable runs are shown in Fig. 1c. No data were excluded on the basis of fMRI data quality.

To examine the relationship between head motion and eye tracking error, we estimated head shift indirectly, as between-participant variation in camera-to-eye distance prevented direct measurement of actual distances from the camera. To approximate head shift in physical units, we conducted post-hoc measurements of each participant's vertical palpebral fissure height (PFH). Thirteen of the original 18 participants were recontacted to obtain PFH in millimeters, and the corresponding PFH in pixels was extracted from their eye tracking videos to compute participant-specific pixel-to-millimeter scale factors. For the remaining five participants, we used the group-mean PFH (mean = 10.73 mm, s.d. = 1.31 mm) to estimate the scale. The estimated head shift (in mm) during the experiment was computed by multiplying this scaling factor by the mean change in pupil coordinates (in pixels) from initial calibration to the validation stage. This procedure provided a relative measure of head shift, which was then used to analyze its relationship with eye tracking error.

**Motion-corrected eye tracking (MoCET)**

To compensate for head motion effects in eye tracking data, we used participants' head motion parameters obtained from fMRI preprocessing. These parameters consist of six degrees of freedom (6 DoF) motions, capturing translations along the $x$, $y$, and $z$ axes as well as rotations around these axes. These parameters, used during the motion correction step, reflect the relative displacement of the head from a reference template image. To ensure that the head motion parameters calculated from the BOLD reference image estimated during fMRIprep preprocessing, aligned with the initial eye tracking calibration at the beginning of the experiment, we re-centered them by subtracting the head motion values of the first volume from all subsequent volumes. This adjustment yielded head motion displacements relative to the initial calibration position, which were then used to correct for head motion-induced error in the eye tracking data and to

generate simulated eye movement under head motion in our computational model simulation.

To mitigate head motion-related error in the eye tracking data, we applied a signal denoising method based on linear regression. For the main analysis, our regressors included the six head motion parameters as well as polynomial terms up to the third degree (cubic). The head motion parameters were upsampled using linear interpolation to match the temporal resolution of the eye tracking data at 60 Hz. Using these combined regressors, we constructed a linear regression model to predict the $x$ and $y$ pupil coordinates. The model predictions were then subtracted from the original pupil coordinates, resulting in motion-corrected pupil coordinates. We applied polynomial detrending using a similar approach to compare eye tracking accuracy across different detrending methods.

**Across run generalization.** To test whether MoCET can correct inter-run eye tracking drift without repeated calibrations, we implemented an across-run correction protocol. For each participant, a single calibration run was selected (the run with the lowest validation error), and a linear regression model was trained to map 6-DoF head motion parameters to pupil displacement. To extend this model across runs, head motion parameters in the other runs were realigned to the same reference volume (i.e., the first volume of the calibration run), enabling prediction of motion-induced drift across separate scan runs. The learned model weights from the calibration run were then directly applied to all other runs without re-fitting.

In our full model, MoCET combines head motion regressors with low-order polynomial terms to capture both motion-induced and non-motion-related error. Since polynomial terms are time-dependent and cannot be reused across runs, we applied across-run MoCET in three steps: (1) apply motion correction using trained weights from the calibration run, and (2) fit new polynomial detrending per run to remove residual drift. We compared across-run MoCET to two baselines: (1) re-centering the gaze origin at the start of each run, mimicking manual drift correction, and (2) re-centering followed by polynomial detrending to reduce slow drift. Performance was assessed using validation error across non-calibration runs. We also evaluated performance separately for verifiable and unusable runs (based on calibration quality) to assess robustness of MoCET to poor or missing calibration data.

**Nonlinear motion-correction.** While MoCET assumes a linear relationship between head motion and drift in eye tracking data, this assumption may not hold under all circumstances. In particular, when head motion is large or abrupt, a simple linear regression model may be insufficient. To evaluate when the linearity assumption in MoCET remains valid and whether more complex, nonlinear models provide added value, we categorized eye tracking runs based on their head motion profiles and compared multiple versions of MoCET under these conditions. 'Large motion' runs were defined as those with a maximum in-plane displacement exceeding a +1 $z$-score threshold. To identify drift-like motion, we computed autocorrelations of each of the six head motion parameters over lags of one to five TRs, then averaged across lags and parameters. Runs with a $z$-scored mean autocorrelation above +1 were classified as drift. In contrast, spike-like motion was characterized by abrupt, transient movements, defined as runs with a $z$-scored mean autocorrelation below −1.

To test whether nonlinear modeling improves performance under different head motion types, we introduced two extended models. MoCET-Large incorporates squared terms and temporal derivatives of the head motion parameters to capture larger or more dynamic shifts. MoCET-Interaction includes pairwise interaction terms between head motion axes to account for cross-dimensional effects. These models were evaluated alongside the original linear approach to assess the benefit of increased model complexity.

To examine the validity of the linear assumption under more extreme conditions, we performed a simulation-based analysis. Because large head movements are difficult to elicit reliably during empirical fMRI sessions, we artificially amplified head motion by scaling the original motion parameters by 20% to 360%. This approach allowed us to systematically test MoCET and its nonlinear variants under a wide range of motion intensities, including cases beyond typical experimental variability.

## Computational simulation of eye movement

**Anthropometric parameters of model simulation.** Anthropometric measurements of adult human heads adopted from previous studies[26–28] were slightly adjusted to create an isotropic, spherical head model for our simulation. For example, three head dimensions—head breadth (ear-to-ear distance), horizontal depth (back of head to nose), and vertical length (chin to top of head)—were averaged to define a single parameter representing the diameter of the model's head sphere. The model's eyeballs are physically attached to this head sphere, such that head movement affects the 3D geometric position of the eyeballs, while each eyeball can rotate independently to simulate gaze toward target locations. As the pupil is attached to the eyeball, its 3D coordinates change in accordance with the eyeball's rotation, maintaining alignment along the direct line from the eyeball center to the target gaze location. Supplementary Fig. 1 provides detailed anthropometric parameters of the head and eyeball used in the computational model.

**Optimization of eye tracking camera parameters.** A virtual eye tracking camera was positioned in front of the model's left eye to capture simulated eye movements. Due to variations in participant head size, eye position, and setup alignment, the direction of the actual camera installed within an MRI scanner may not be completely perpendicular to the eyeball, often resulting in slight tilts. To replicate realistic eye tracking as closely as possible, we customized the virtual camera's direction, roll angle, and distance from the model's eyeball, calibrating these parameters to match participant-specific eye tracking data. This was achieved by simulating a range of configurations for four camera parameters. For the camera's orientation, the baseline direction was set to face the model's eyeball directly; yaw and pitch angles were then varied by up to ±5 degrees along both horizontal and vertical axes. The camera's roll angle was parameterized within a range of −90 to +90 degrees. Finally, the camera-to-eyeball distance was adjusted from 4.5 to 15 cm to account for natural variations in participant head and eye positioning relative to the fixed camera position in the head coil. This setup created ~36,000 unique combinations of four camera parameters (yaw, pitch, roll, and distance).

To determine the optimal camera parameters for each participant, we simulated the model's gaze directed toward 12 calibration points on a screen, producing twelve 2D pupil locations as captured by the virtual eye tracking camera. These simulated pupil locations were then compared to the participant's actual pupil coordinates recorded for the same 12 calibration points. For each of the 36,000 parameter combinations, we calculated the discrepancy between the model's and participant's pupil locations, and selected the optimal parameters for each participant through a grid search that minimized this difference. The optimized camera parameters averaged across participants and examples of parameters for individual participants are shown in Supplementary Fig. 1.

**Incorporating head motion parameters.** At each step of the simulation, we constructed a 3D rotational transformation matrix and a 3D translational transformation matrix from head motion parameters. These matrices were then applied to the coordinates of the model's head and attached eyeball, enabling our simulation to precisely reproduce participants' head motions in the simulation environment.

**Analysis of simulated gaze.** The model's eye movement was recorded as a sequence of images using a virtual eye tracking camera. Ray casting was employed to generate 2D projections of 3D objects (head, eyeball, and pupil) by tracing rays from the virtual camera until they intersected with object surfaces, determining their visible positions[55]. For visualization (e.g., Fig. 2 and Supplementary Fig. 1), a light source and the Blinn-Phong reflection model[56,57] were used to render realistic colors and shadows. However, for efficient simulation and accurate pupil tracking, rendering was simplified by assigning fixed colors to objects without light reflection, ensuring only the closest intersecting object was represented. This optimized both simulation speed and accuracy while preserving essential pupil location details.

Simulated eye tracking data were preprocessed and analyzed using the same procedure as the human eye tracking data. To examine the relationship between the magnitude of head motion and eye tracking error, we calculated the extent of head shift occurring within the $p_x$-$p_y$ plane, from initial calibration to final validation. While the magnitude of head shift in human eye tracking data was estimated relative to the assumed pupil size, the model simulation allowed for the precise measurement of the actual physical distance of head shift in 3D space.

**Permuted random head motion simulation.** To assess the statistical significance of the relationship between head motion and pupil coordinates, we generated permuted random head motion parameters. The randomization was performed using a phase-shuffling approach, which preserves the power spectrum and temporal autocorrelation of the original head motion time series. Specifically, the Fourier transform of each head motion parameter was computed, and the phase components were randomly shuffled while keeping the amplitude spectrum intact. The inverse Fourier transform was then applied to reconstruct the randomized head motion time series.

This procedure ensures that the randomized head motion retains the same temporal structure as the original data but removes any relationships with the actual experimental conditions. We applied this method independently to each of the six head motion parameters. These randomized head motion parameters were then used to simulate pupil movements in the geometry-based eyeball model, producing a null distribution of gaze similarity metrics. Statistical significance was determined by comparing the actual simulation results with the null distribution obtained from 100 permutations for each dataset. To evaluate overall significance across the dataset, we aggregated individual $p$-values using Fisher's method[29], providing a robust summary statistic for group-level analyzes.

## Predicting behaviors from gaze patterns

To evaluate the predictive power of drift-corrected gaze data for participant behavior, we analyzed data from the Minecraft-based video game task. In this task, participants sequentially built new fences or removed existing fences within a square arena to create a shortcut from their starting locations to a designated destination. Each participant's decisions and actions were logged in-game, allowing us to identify their next target block (fence) based on their strategy. We used this behavioral dataset to compare the predictive performance of eye tracking data corrected using MoCET, linear detrending, polynomial detrending, and uncorrected data.

The in-game log file, recorded at 20 frames per second, included the 3D physical coordinates of each target block in the Minecraft world. To determine the target's corresponding location on the screen, we used the participant's in-game camera position, including their locations, yaw (horizontal rotation), and pitch (vertical rotation). These parameters were used to project the 3D world coordinates of the next target block onto 2D screen coordinates relative to the participant's FOV. This process was repeated for each frame to account for participant movement or camera adjustments during the task.

For each moment, the next target block was identified based on the in-game log, and the distance between the participant's gaze and the 2D screen coordinates of the target was calculated for a 10-s period preceding the action (building or removing a block). A "hit" was defined as a gaze position falling within a 2-degree visual angle from the target's location on the screen. The hit rate of each action was then calculated as the percentage of frames within the 10-s window (i.e., 200 frames) in which a hit occurred. Chance-level performance was estimated based on the probability of a random gaze location falling within a 2-degree visual angle around the target. Specifically, the circular area corresponding to a 2-degree visual angle ($\pi r^2$) was divided by the total screen area, assuming a uniform distribution of gaze locations across the screen. This provided a baseline probability of a random gaze intersecting the target. The overall hit rate for each participant was averaged across all actions. Statistical comparisons of hit rates between the different eye tracking data correction methods were performed using paired $t$-tests.

### Visual field mapping from free-viewing visual stimuli

To evaluate the accuracy of retinotopic mapping using free-viewing visual stimuli, we used drift-corrected gaze data for pRF modeling[40,41]. The free-viewing stimuli consisted of the Minecraft video game display as participants played the task. Gaze data corrected using MoCET, polynomial detrending, linear detrending, and uncorrected data were used to generate gaze-centered stimulus representations and assess their impact on retinotopic mapping in the visual cortex.

**Stimulus representation and spatiotemporal local contrast**. For each frame of the video game display (original resolution: $1600 \times 1000$ pixels), we generated gaze-centered spatiotemporal local contrast by extracting stimulus patches relative to the participant's gaze. To account for potential gaze coordinates outside the visible screen, the original frame was padded with gray borders, creating an expanded frame of $4800 \times 4800$ pixels. Using the gaze coordinates, a $3200 \times 3200$ pixels patch was cropped from the padded image, ensuring the gaze remained at the center of the extracted stimulus.

The cropped patch was then resized to a resolution of $320 \times 320$ pixels. To compute spatiotemporal local contrast, each upsampled frame was divided into small 3D patches of $5 \times 5$ pixels spatially and 1.6 s temporally (48 frames). Standard deviation values were calculated within each 3D patch, resulting in a spatiotemporal local contrast representation with a resolution of $64 \times 64 \times 510$. This stimulus energy signal served as the input to the pRF model[39].

**pRF model fitting and analysis**. The pRF model estimated polar angle representations in the visual cortex by fitting a Gaussian receptive field centered on the predicted gaze location to voxelwise BOLD responses. The visual cortex mask was extracted from a resting-state functional atlas[58]. Model fitting utilized the spatiotemporal local contrast signal derived from gaze-corrected free-viewing stimuli as the input and voxelwise time-series data as the output. The Gaussian receptive field model predicted neural responses to the spatial location of stimuli, enabling the reconstruction of polar angle maps. Model performance was quantified by calculating angular errors between the estimated polar angles and canonical polar angles for ten cortical landmarks, including the calcarine sulcus and V1-V2 and V2-V3 boundaries (dorsal and ventral) in both hemispheres. Cortical landmarks were identified using retinotopic ROIs from the HCP dataset[49], where key reference vertices (5–10 per landmark) were manually selected along the cortical boundaries. A geodesic pathfinding algorithm was then used to interpolate and connect these reference vertices, forming a continuous representation of each landmark. The extracted geodesic lines were used to derive ground-truth polar angles for evaluating model accuracy.

### Training and validation of MR-based eye tracking models

To train and evaluate the MR-based DeepMReye models[21], we first extracted eyeball voxel masks from the fMRI data. The masks were generated using the protocol described in the original DeepMReye reference, and their quality was manually inspected by researchers to ensure proper delineation of the eyeball regions. Three MR-based eye tracking models were compared: the *Pretrained* model, which utilized the original pre-trained DeepMReye weights without modification; the *Finetuned* model, where the pre-trained weights were further trained using our dataset to adapt to the current experimental settings; and the *Scratch* model, which was trained from scratch using only the calibration data from this study.

For the initial calibration stage, 24 volumes corresponding to the presentation of sequential calibration points were used to train the *Finetuned* and *Scratch* models. The calibration points appeared one per volume, providing labeled data for training. Most of the model training parameters, including the neural network architecture, loss function weights, batch size, and augmentation settings, were retained from the original configuration. However, specific adjustments were made to the number of training epochs and the learning rate decay schedule to ensure optimal model performance on the calibration data. These adjustments allowed the model to achieve sufficient convergence as assessed by the loss history during training.

After training, the models were applied to predict gaze coordinates across the entire fMRI experiment. The MR-based models produced one predicted gaze coordinate per fMRI volume. Performance validation was conducted for both the calibration and validation stages. During the validation stage, 12 gaze coordinates were predicted for the 12 validation points presented sequentially. Performance metrics, including gaze inaccuracy (Euclidean error) and directional accuracy (correlation with the ground truth), were calculated in the same manner as for camera-based eye tracking.

### Statistics and reproducibility

All statistical analyzes followed the procedures described in the Methods. The relationship between head motion and pupil coordinates was assessed using simulation-based permutation testing, in which phase-shuffled head-motion parameters generated null distributions for evaluating significance. Eye tracking accuracy, behavioral prediction accuracy, and retinotopic mapping accuracy were compared across methods using paired $t$-tests. Sample sizes were based on prior naturalistic fMRI studies and matched common practice; no statistical method was used to predetermine sample size. Runs were categorized as verifiable, usable, or unusable based on predefined eye tracking accuracy thresholds and only runs with unusable calibration were excluded from analyzes that required accurate within-run calibration. The experiments were not randomized, and investigators were not blinded to allocation during data collection or analysis.

### Reporting summary

Further information on research design is available in the Nature Portfolio Reporting Summary linked to this article.

## Data availability

The eye tracking data and head motion parameters are available at Zenodo with the following link https://zenodo.org/records/17089244. Source data are provided with this paper.

## Code availability

The analysis scripts used in this study, along with the MoCET Python package, are publicly available on GitHub (https://github.com/jwparks/mocet), with a fixed version archived on Zenodo[59]. The package can be installed via PyPI using the command: 'pip install mocet'.

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

## Acknowledgements

We would like to thank Tianjiao Zhang and Jack L. Gallant for their initial development of eye tracking software; Boohee Choi, Kyubeen Ahn, Seoyu Kim and Sunhyun Min for technical support in fMRI and eye tracking data collection; Jiun Choi and Jisu Ro for providing datasets used in the replication analyses; Sanghum Woo, Kyeong-Jin Tark, and Joon-Yeol Lee for their assistance with behavioral eye tracking data collection. This work was supported by NIH grant R01EY034118 to K.N.K., the National Research Foundation of Korea (RS-2024-00348130, RS-2025-02304581) to W.M.S., and the Fourth Stage of Brain Korea 21 Project (S-2023-0794-000) to W.M.S. Authors J.P., J.Y.J., and R.K. were supported by funding from the Institute for Basic Science (IBS-R015-D2).

## Author contributions

J.P., K.N.K., and W.M.S. conceptualized and designed the research. J.Y.J. developed the eye tracking setup tailored for the 7T MRI scanner. J.P., J.Y.J., R.K., K.N.K., and W.M.S. conducted the research. J.P. developed the simulation model. J.P. analyzed the eye tracking data. J.P. and R.K. analyzed the neuroimaging data. J.P. drafted the initial version of the manuscript, and J.P., J.Y.J., R.K., K.N.K., and W.M.S. collaboratively revised and edited the manuscript.

## Competing interests

The authors declare no competing interests.
