## [Transparent Peer Review file · Nature Communications]

Motion-corrected eye tracking improves gaze accuracy during visual fMRI experiments

Corresponding Author: Dr Won Mok Shim

Version 0:

Reviewer comments:

Reviewer #1

(Remarks to the Author)
Summary

MoCET is a retrospective motion correction framework for camera based eye tracking in fMRI. It combines six degree of freedom (6 DoF) head motion parameters with cubic polynomial regressors to compensate gaze drift. Across both simulated and empirical data, the method exceeds traditional detrending and an MR based decoder (DeepMReye) on gaze precision, behavioural prediction, and retinotopic mapping.

After a detailed evaluation, I find the study's methodological rigor, open data sharing, and clear demonstration of MoCET's advantages highly commendable; with only relatively small additions—chiefly clarifying the linear-model limits, refining the pupil-scaling assumption, and expanding the comparison to a non-linear baseline—I believe the paper can be strengthened, and therefore I recommend acceptance subject to minor revision.

Major Comments

1. Linearity assumption

The model presumes a linear mapping between 6 DoF head motion and image plane drift, yet non linear effects can arise during large translations or oblique camera angles. Please implement at least one non linear alternative—such as a model with second order interaction terms or a kernel based regression—and compare it with the linear model using information criteria or cross validated error. In parallel, quantify the motion range (e.g., > 5 mm translation or > 5° rotation/tilt) at which the linear fit becomes inadequate. This can be done by mining high motion volumes already present or by collecting a short auxiliary scan in which participants deliberately move more.

2. Pupil scale factor

The manuscript assumes a uniform 5 mm pupil diameter when converting pixel displacement to millimetres; in practice, inter-individual variation can be ± 1 mm or more, producing scaling errors of 20 % or greater. A straightforward remedy would be to place a small fiducial of known size (e.g. a 3 mm circular “target” sticker) on each participant's cheek pad or forehead within the camera's field of view. Because the sticker's pixel width is constant across frames, it provides an individualised millimetre-per-pixel scale that (i) eliminates the need to guess pupil size, (ii) remains valid even if pupil diameter changes with arousal, and (iii) would allow MoCET, in principle, to be extended to pupillometry analyses that require absolute size accuracy.

3. Temporal resolution mismatch

Because fMRI motion parameters are sampled every TR (1.6 s) while eye tracking is acquired at 60 Hz, the two signals differ in temporal resolution. Beyond the simulations already provided, please demonstrate how down sampling (i.e., subsampling the 60 Hz eye-tracking signal to the fMRI TR of 1.6 s) the eye tracking data to the fMRI rate influences correction fidelity. A brief supplementary figure comparing full rate and down sampled corrections would clarify the practical impact of this mismatch.

4. DeepMReye calibration duration

DeepMReye's weaker performance may stem from the shortened fixation duration (1 TR) used during calibration—substantially briefer than the 4 s dwell time in the original protocol. Stable fixation is critical for accurate training; this limitation should be explicitly acknowledged. Please also state whether hyper-parameter tuning or additional fine tuning was attempted for these abbreviated fixations, and briefly justify the conceptual value of comparing a camera based correction (MoCET) with a voxel based decoder (DeepMReye).

5. Sample size rationale and run variability

The manuscript outlines recruitment and exclusion criteria but does not explain how the final sample sizes ($n = 20$ main; $n = 18$ pRF) were determined. Indicate whether these numbers were based on power analysis or prior work. Because valid runs per participant varied widely, explain how this variability was accommodated (e.g., weighting, run thresholds). Finally, clarify why 13 participants completed two sessions while eight completed one—was this a design decision, dropout, or exclusion? Note any fMRI specific exclusions (e.g., excessive motion) that shaped the final sample.

6. To illustrate the generalisability of MoCET, I recommend two supplementary transfer-tests. First, assess "cross-participant" transferability by applying each participant's regression coefficients to all other participants' eye-tracking data, then report the resulting gaze error relative to (i) participant-specific MoCET, (ii) polynomial detrending, and (iii) the uncorrected signal; this will quantify how strongly the model depends on individual camera-to-eye geometry. Second, evaluate "within-participant", across-session stability by re-using coefficients estimated in session 1 to correct session 2 (acquired on a different day), and compare their performance with coefficients re-estimated from session 2's own calibration; if cross-day error increases only marginally, it would highlight MoCET's practical advantage of shortening or omitting calibration in follow-up scans.

Minor Comment

The terms "drift," "error," and "noise" are used interchangeably throughout the manuscript. Please define them consistently at the outset and use the preferred terms thereafter.

(Remarks on code availability)

Given the limited review window, I did not have capacity to inspect the submitted code line-by-line; instead, I concentrated on the conceptual soundness of the design, statistics, and interpretation. Because my comments request new analyses (e.g., a non-linear correction model and a sampling-rate sensitivity test), the code will necessarily change in the revision, at which point a full code audit would be more appropriate.

Reviewer #2

(Remarks to the Author)

(Remarks on code availability)

Reviewer #3

(Remarks to the Author)

This work presents a method for motion correcting eye tracking data collected within an MRI scanner using MR-derived motion parameters (MoCET). The authors document the challenges of in-scanner eye tracking by showing that eye tracking error rates increase over the course of a single MRI run. They hypothesize that the errors are due to in-plane head movement shifts and indeed find a high correlation between the validation eye tracking error and the in-plane head movement as measured with standard MRI head tracking techniques. They further establish the strong relationship between in-plane head movement and eye tracking errors using a computational simulation of an eyeball being eye tracked, and again find that eye tracker validation errors are a function of the in-plane head movement in their simulation. To correct for head movement in the eye tracking data, the authors adapt standard fMRI motion-correction procedures for eye tracking by regressing out the 6DoF head position parameters from the eye tracking data, finding that this reduces the validation error down to acceptable levels and outperforms other standard approaches for denoising eye tracking data. Next, they test MoCET in the context of two different tasks: predicting human behavior during an in-scanner freeviewing task and deriving retinotopic pRF maps, finding that MoCET-corrected data leads to the best performance on both tasks. Finally, they compare their method to DeepMReye, an MR-based method for decoding eye positions directly from the MRI data of the eye, and find that MoCET outperforms DeepMReye.

Overall, this is extremely high quality work both in execution and presentation. The methodological problem of collecting usable concurrent eye tracking data during MRI sessions is one that has been persistent throughout my own work and has stymied experiments over the years. I am very pleased to see an elegant approach that mostly solves the problem. The authors thoroughly characterize the problem and the success of their correction procedure. The paper is clearly written. I have one analysis request and several small comments.

Main comment

- I am curious how the drift in eye tracker error between calibration and validation, and the reduction in error from MoCET, compare to standard within-run drift you see in standard eye tracking setups outside MRI scanners with head rests and/or bite bars.

Relatedly, I'm curious for the behavioral prediction analysis what the prediction performance would be for standard eye tracking setups.

- This could be addressed by repeating the in-scanner experiments on a standard eye tracking setup. I'm not familiar with the brand of eye tracker used for the in-scanner experiments, but I don't think the out-of-scanner tracker would necessarily need to exactly match the in-scanner tracker. Ideally it's a setup with a chin rest for head stabilization, as this in my experience is the most common setup for out-of-scanner head stabilization in eye tracking.

- The benefits of this experiment are twofold.

- First, it provides a reference frame for the in-scanner results. In the section "Impact of head motion on gaze tracking accuracy" and Figure 1, showing the calibration and validation error for an out-of-scanner eye tracker would provide a useful anchor for the scale of the problem for in-scanner eye tracking by showing the magnitude of the problem for in-scanner setups vs out-of-scanner setups. In the section "Compensating for head movement with Motion-Corrected Eye Tracking" and Figure 4a, the validation error for a standard out-of-scanner eye tracker would provide a floor for the error that indicates the degree of success. If MoCET reduces the validation error such that it is statistically the same as the validation error for an out-of-scanner eye tracking setup, that provides strong evidence that MoCET has functionally solved the targeted problem.

- Second, it provides a reference frame for the behavioral prediction performance in the section "MoCET enables high-precision eye tracking during free-viewing experiments" and Figure 4B. As currently presented, it's clear that MoCET allows some prediction performance improvements over standard in-scanner eye tracking correction approaches, but I'm not sure whether the 15% hit rate is in line with what should be expected from standard head stabilized eye tracking.

Minor comments

- Page 23, "Eye tracking accuracy", Paragraph 2 says that all data w/ calibration/validation errors of >1degree were excluded from subsequent analysis "to exclude errors stemming from fixation instability or individual gaze variability, thereby ensuring that any remaining inaccuracies were not simply due to inconsistent eye movements." I'm not sure how this could be the case, given that much larger validation errors are reported in the paper. In the first Results section and Figure 1, the reported validation error is 7.093 degrees and in Figure 4 the uncorrected validation errors are in the range of 4-7 degrees. Please clarify if I misunderstood and adjust the text so the exclusion criteria are clear and consistent with the main text.

- How big are the stimuli in the fMRI experiments in degrees of visual angle? I couldn't find this anywhere in the main text or the methods, and this is helpful for determining the size of the eye tracker errors (which are already reported in degrees) relative to the stimuli.

- DeepMReye comparison. Does the fMRI data need to be normalized to MNI space for the DeepMReye model to run? Since the analyses are within-participant, normalizing to MNI is not essential for group-level comparison, and DeepMReye may show better performance in each participants native space.

(Remarks on code availability)

Reviewer #4

(Remarks to the Author)

This paper by Park et al. presents a methodologically strong solution to correcting eye-movement measurement drift with head-motion data in fMRI. The proposed method, Motion-Corrected Eye Tracking (MoCET), that retrospectively corrects for motion-induced drift by leveraging the 6-degree-of-freedom head motion parameters that are a standard output of fMRI preprocessing. This approach makes a significant and accessible contribution to the field.

Strengths

By capitalizing on data already generated in every standard fMRI pipeline, the method eliminates the need for additional hardware or complex setups. This lowers the barrier to high-quality eye-tracking research, making it accessible to a much broader community of scientists. The ability to apply it retrospectively is a major advantage, potentially salvaging vast amounts of existing data.

The authors employ an elegant dual-validation strategy that uses computational simulations effectively isolate the causal effects of head motion, confirming it as the primary source of drift. This is powerfully complemented by empirical data from human participants, where MoCET's performance is demonstrated across multiple, functionally relevant metrics. Further remarks below.

1. Although the method is itself general and practical, the acquisition protocol perhaps less so. One would have liked to see generalization across more than one 7T dataset, one of which involved a fairly unusual task (Minecraft).

2. Head motion can be quite variable across subjects. There may be a relationship between the characteristics of head-motion across subject (more or less high-frequency events, variation is predominant motion type, etc.) that could affect the effectiveness of MoCET. If head-motion is more "drift-like" its will be more similar to the polynomial regressor approach. But if motion is more spiky, then these methods will diverge, as nonlinearities may be more important (or that robust regression might yield better results). In general more discussion of individual variability in head-motion would be desirable; especially

in light of point 1 (generalization); for example, where different patient populations or age-groups might have different characteristic head-motion profiles.

3. Related to above: there is an inherent discrepancy between the high sampling rate of eye-tracking (e.g., 60 Hz) and the low sampling rate of fMRI-derived motion estimates (~0.5-1 Hz). Although the authors argue convincingly that head motion is dominated by low-frequency components, fast, transient movements (e.g., respiration-related) may not be perfectly corrected and could partially alias into the residual noise.

Overall, MoCET is a well-motivated and rigorously evaluated solution that advances the fidelity of eye tracking in the challenging MRI environment. The paper is technically sound, the validation is strong, and the open-source implementation significantly enhances its potential impact.

(Remarks on code availability)

Version 1:

Reviewer comments:

Reviewer #1

(Remarks to the Author)

I have no further substantive requests. I accept the revisions and thank you again for your thorough work.

(Remarks on code availability)

Reviewer #2

(Remarks to the Author)

(Remarks on code availability)

Reviewer #3

(Remarks to the Author)

The updates and response address my critiques of the original manuscript, and I'm happy to recommend publication. Thank you to the authors for extremely thorough responses to the reviews and completing multiple new experiments for the updated version. This is excellent work overall, and I look forward to using MoCET in my own research.

(Remarks on code availability)

Response to *Nature Communications* Reviewers

Motion-corrected eye tracking improves gaze accuracy during visual fMRI experiments

Jiwoong Park, Jae Young Jeon, Royoung Kim, Kendrick N. Kay, Won Mok Shim

We sincerely thank the reviewers for their thoughtful and constructive feedback, which has substantially improved the clarity, rigor, and scope of our manuscript. In response, we have made substantial revisions to both the main text and supplementary materials, addressing each of the reviewers' comments in detail.

A key improvement in this revision is the expanded validation of MoCET's performance and applicability. Specifically:

- **We implemented two nonlinear variants of MoCET to evaluate whether the linear model sufficiently captures the mapping between head motion and eye tracking drift.** These variants were compared using both simulations and human data. Results showed no significant advantage of the nonlinear models, supporting the parsimonious linear assumption under typical scanning conditions.
- **We conducted systematic simulations to evaluate the impact of large head motion,** identifying the range where linear assumptions may begin to degrade and where nonlinear models offer marginal gains.
- **We added an across-run variant of MoCET, which enables calibration transfer within the same scan session.** This method demonstrated high correction accuracy without requiring repeated calibrations per run, offering a practical solution for in-scanner eye tracking experiments.
- To provide a baseline for comparison, **we conducted behavioral eye tracking experiments outside the MRI scanner** using a standard laboratory setup. These results show that in-scanner eye tracking data, corrected by MoCET, were statistically comparable to out-of-scanner behavioral eye tracking in both gaze accuracy and behavioral prediction performance, providing a critical reference.
- **We added results from an independent replication dataset collected using a different free-viewing task and scanner,** demonstrating that MoCET generalizes across tasks and imaging environments.
- **We addressed the temporal resolution mismatch between motion parameters and eye tracking signals** by evaluating the impact of downsampling eye data to fMRI sampling rates. Results confirm MoCET's robustness under realistic conditions.

- **We clarified the data inclusion criteria and the distribution of usable data** across participants and runs, as requested, improving transparency about sample size and data quality thresholds.
- The **MoCET software package has been updated to support** widely used eye tracking data formats (e.g., EDF files from EyeLink) and motion parameters from alternative preprocessing tools (e.g., FSL). All new features and analyses are reflected in the revised codebase.
- For clarity, **we have revised terminology throughout the manuscript.**

These revisions address the reviewers' concerns comprehensively and strengthen the contribution of MoCET as a robust, practical solution for in-scanner eye tracking during fMRI experiments. We appreciate the reviewers' insightful comments and hope that the revised manuscript meets the journal's standards for publication.

Please see below for a point-by-point response to the reviewers' comments. All revised text is marked in **red** in the manuscript for ease of review.

Reviewer #1

1. **Linearity assumption: The model presumes a linear mapping between 6 DoF head motion and image plane drift, yet non linear effects can arise during large translations or oblique camera angles. Please implement at least one non linear alternative—such as a model with second order interaction terms or a kernel based regression—and compare it with the linear model using information criteria or cross validated error. In parallel, quantify the motion range (e.g., > 5 mm translation or > 5° rotation/tilt) at which the linear fit becomes inadequate. This can be done by mining high motion volumes already present or by collecting a short auxiliary scan in which participants deliberately move more.**

Response: We thank the reviewer for highlighting this important issue regarding the linearity assumption in MoCET. While MoCET models a linear relationship between 6 DoF head motion parameters and pupil displacement on the image plane, we agree that nonlinear effects can emerge during large head translations or oblique camera angles due to the spherical geometry of the head and optical distortions (Please see also Reviewer #4's comment 2 for a similar concern).

To address this concern, **we implemented and evaluated two nonlinear extensions of MoCET:**

- **MoCET-Large**, which includes 24 regressors, combining the original 6 DoF motion parameters with their temporal derivatives, squared terms, and squared derivatives, along with polynomial terms up to cubic order.
- **MoCET-Interaction**, which adds all pairwise second-order interaction terms among the 6 motion parameters, in addition to the original set and the same polynomial terms.

We classified empirical head motion into three types: large motion, spike-like motion, and drift-like motion, based on in-plane displacement and autocorrelation measures (Supplementary Figs.

3a and 3b). Using these categories, we compared correction accuracy between the linear model and the two nonlinear variants (Supplementary Fig. 3c). Across all conditions, we found no significant differences in correction accuracy between the linear and nonlinear models.

To further quantify the conditions under which the linear approximation might break down, we conducted a simulation in which head motion parameters were amplified by scaling their original values by 20% to 360% (Supplementary Fig. 3d). This allowed us to simulate scenarios with moderate to extreme head motion. Results showed that the standard linear MoCET consistently outperformed the nonlinear models up to approximately 140% scaling. MoCET-Large began to show advantages only when motion exceeded 200%, while MoCET-Interaction did not offer reliable improvements and was more susceptible to overfitting.

These results suggest that the linear assumption used in MoCET is well-suited for typical fMRI scanning conditions, where head motion is generally mild to moderate. While we agree that collecting auxiliary scans with deliberate motion could offer additional insight, our simulations approximate a wide range of realistic motion levels observed in practice.

Finally, to support broader use cases, **we have included two nonlinear extensions of MoCET as optional modules in our open-source Python implementation**, enabling users to apply these models when higher-order motion effects are expected.

“To further test the linear assumption, we evaluated the nonlinearity in the relationship between head motion and pupil coordinates across different types of head motion. We found that the linear and nonlinear variants of MoCET performed comparably across motion categories. Additional simulations with artificially amplified motion indicated that the linear MoCET model remained effective at motion levels typical of fMRI studies, while nonlinear models offered only limited additional benefit under extreme motion (for details, see Supplementary Fig. 3 and Methods)” —Results (page 11)

“MoCET assumes a linear relationship between head motion and gaze error, which may become inadequate in cases of extreme head movement or when the eye tracking camera is installed at a steep tilt relative to the gaze direction. In such conditions, nonlinear distortions in pupil coordinates can introduce errors not fully accounted for by linear regression. However, our empirical evaluation across diverse head motion types, including large, spike-like, and drift-like movements, showed that the linear MoCET model performed as well as more complex nonlinear variants. Furthermore, simulations with systematically amplified motion confirmed that linear MoCET remains effective across a broad range of motion levels. Nonlinear extensions provided improvements only under extreme head motion conditions, which are rarely encountered in empirical experiments (Supplementary Fig. 3). These findings support the validity of the linear assumption in typical fMRI settings, where head motion is often constrained by stabilization devices such as foam padding, minimizing extreme deviations that could introduce nonlinear distortions. To accommodate potential edge cases, we also provide these nonlinear MoCET variants in our Python package, allowing users to apply more flexible correction models when needed.” —Discussion (page 21)

“While MoCET assumes a linear relationship between head motion and drift in eye tracking data, this assumption may not hold under all circumstances. In particular, when head motion is

large or abrupt, a simple linear regression model may be insufficient. To evaluate when the linearity assumption in MoCET remains valid and whether more complex, nonlinear models provide added value, we categorized eye tracking runs based on their head motion profiles and compared multiple versions of MoCET under these conditions. ‘Large motion’ runs were defined as those with a maximum in-plane displacement exceeding a +1 z-score threshold. To identify drift-like motion, we computed autocorrelations of each of the six head motion parameters over lags of one to five TRs, then averaged across lags and parameters. Runs with a z-scored mean autocorrelation above +1 were classified as drift. In contrast, spike-like motion was characterized by abrupt, transient movements, defined as runs with a z-scored mean autocorrelation below -1 .

To test whether nonlinear modeling improves performance under different head motion types, we introduced two extended models. MoCET-Large incorporates squared terms and temporal derivatives of the head motion parameters to capture larger or more dynamic shifts. MoCET-Interaction includes pairwise interaction terms between head motion axes to account for cross-dimensional effects. These models were evaluated alongside the original linear approach to assess the benefit of increased model complexity.

To examine the validity of the linear assumption under more extreme conditions, we performed a simulation-based analysis. Because large head movements are difficult to elicit reliably during empirical fMRI sessions, we artificially amplified head motion by scaling the original motion parameters by 20% to 360%. This approach allowed us to systematically test MoCET and its nonlinear variants under a wide range of motion intensities, including cases beyond typical experimental variability.” —Methods (page 29)

—Supplementary Figure 3 (page 44)

- 2. Pupil scale factor:** The manuscript assumes a uniform 5 mm pupil diameter when converting pixel displacement to millimetres; in practice, inter-individual variation can be ± 1 mm or more, producing scaling errors of 20 % or greater. A straightforward remedy would be to place a small fiducial of known size (e.g. a 3 mm circular “target” sticker) on each participant’s cheek pad or forehead within the camera’s field of view. Because the sticker’s pixel width is constant across frames, it provides an individualised millimetre-per-pixel scale that (i) eliminates the need to guess pupil size, (ii) remains valid even if pupil diameter changes with arousal, and (iii) would allow MoCET, in principle, to be extended to pupillometry analyses that require absolute size accuracy.

Response: We thank the reviewer for raising this important point regarding the assumptions underlying our pupil size scaling. We clarify that MoCET’s primary contribution is to correct pupil position, not pupil size. The assumed pupil diameter (5 mm) was used only to convert displacement values from pixels (px) to millimeters (mm) for figure visualization (e.g., Figs. 1d and 2c).

However, we agree that a realistic, individualized millimeters-per-pixel scaling factor would improve visualization accuracy and reliability and facilitate future pupillometry applications.

By contrast with behavioral eye tracking setups, the field of view (FOV) in MRI-compatible eye tracking systems is narrow and typically restricted to the immediate peri-ocular region (See Fig. 1). This constraint makes it impractical to place a fiducial marker on the cheek or forehead without interfering with pupil detection.

To address this limitation, **we conducted an additional post-hoc measurement study. We re-contacted 13 out of the original 18 participants and measured vertical palpebral fissure height (PFH).** We then extracted the corresponding PFH in pixels from the eye-tracking videos to calculate participant-specific pixel-to-millimeter scale factors.

We used these participant-specific scale factors to revise figures reporting mm units. The updated plots show improved accuracy and robustness, supporting the reviewer’s suggestion to adopt more realistic scaling.

For the remaining five participants who did not return for re-measurement, we used the group mean PFH from the 13 participants to estimate the scale. We thank the reviewer for raising this point; it meaningfully improved the accuracy of our visualizations and enabled more precise size-based analyses in future work.

“To examine the relationship between head motion and eye tracking error, we estimated head shift indirectly, as between-participant variation in camera-to-eye distance prevented direct measurement of actual distances from the camera. To approximate head shift in physical units, we conducted post-hoc measurements of each participant’s vertical palpebral fissure height (PFH). Thirteen of the original 18 participants were recontacted to obtain PFH in millimeters, and the corresponding PFH in pixels was extracted from their eye tracking videos to compute participant-specific pixel-to-millimeter scale factors. For the remaining five participants, we used the group-mean PFH (mean = 10.73 mm, s.d. = 1.31 mm) to estimate the scale. The estimated head shift (in mm) during the experiment was computed by multiplying this scaling factor by the mean change in pupil coordinates (in pixels) from initial calibration to the validation stage. This procedure provided a relative measure of head shift, which was then used to analyze its relationship with eye tracking error.”—Methods (page 27)

- 3. Temporal resolution mismatch: Because fMRI motion parameters are sampled every TR (1.6 s) while eye tracking is acquired at 60 Hz, the two signals differ in temporal resolution. Beyond the simulations already provided, please demonstrate how down sampling (i.e., subsampling the 60 Hz eye-tracking signal to the fMRI TR of 1.6 s) the eye tracking data**

to the fMRI rate influences correction fidelity. A brief supplementary figure comparing full rate and down sampled corrections would clarify the practical impact of this mismatch.

Response: We thank the reviewer for raising this concern. While MoCET relies on motion estimates at a lower sampling rate, its core assumption is that head motion primarily contributes low-frequency components to gaze drift, thus allowing low-frequency motion regressors to capture and correct motion-related artifacts without attenuating rapid eye movements (see also Reviewer #4’s comment 3).

To empirically test the impact of this time-resolution mismatch, **we systematically downsampled the original 60 Hz eye tracking data to lower sampling rates (down to 1 Hz).** For each downsampled dataset, we applied MoCET as well as linear and polynomial detrending, and then evaluated eye tracking accuracy. As shown in Supplementary Fig. 7, **MoCET maintained robust correction performance down to 10 Hz, and outperformed standard detrending methods even at extremely low sampling rates (e.g., 1 Hz).** These results indicate that MoCET is robust to temporal resolution mismatches and remains effective even when gaze signals are corrected at the sampling rate of fMRI motion estimates.

“To address this concern empirically, we systematically downsampled eye tracking data from 60 Hz to lower sampling rates and tested MoCET’s performance. The results demonstrate that MoCET’s accuracy remains stable down to 10 Hz, and even under extreme downsampling (to 1 Hz), it outperforms conventional detrending approaches (Supplementary Fig. 7).” —Results (page 21)

—Supplementary Figure 7 (page 48)

4. **DeepMReye calibration duration:** DeepMReye’s weaker performance may stem from the shortened fixation duration (1 TR) used during calibration—substantially briefer than the 4 s dwell time in the original protocol. Stable fixation is critical for accurate training; this limitation should be explicitly acknowledged. Please also state whether hyper-parameter tuning or additional fine tuning was attempted for these abbreviated fixations, and briefly

justify the conceptual value of comparing a camera based correction (MoCET) with a voxel based decoder (DeepMReye).

Response: First, we acknowledge that our calibration protocol used a shorter fixation duration (1 TR, 1.6 s) per target than the 4-second duration (5 TR) employed in the original DeepMReye protocol. This shorter duration may increase variability in training samples due to unstable fixations. However, both the *fine-tuned* and *scratch* DeepMReye models showed strong performance during training, indicating that the reduced fixation time did not substantially impair model fitting. As shown in Fig. 6b, these models accurately captured the calibration data, suggesting that our hyperparameters and protocol were sufficient for effective learning under practical constraints.

We also believe it is important to note the experimental trade-offs. Implementing longer fixation durations would significantly increase the calibration burden, especially with larger numbers of targets. In our case, extending to 5 TRs (8 seconds) per target across 24 locations would require over 3 minutes solely for calibration, which is often impractical in many fMRI protocols. **If MR-based eye tracking requires this level of calibration time to achieve competitive performance, that represents an important limitation of such methods.**

Finally, we agree that the conceptual motivation for this comparison merited further clarification. In the revised manuscript, we have expanded the rationale and emphasized the conceptual value of the comparison.

“However, because MR-based methods are constrained by the spatial and temporal resolution of fMRI, it remains unclear whether they can substitute camera-based systems in applications requiring precise, moment-to-moment gaze tracking.”—Results (page 16)

- 5. Sample size rationale and run variability: The manuscript outlines recruitment and exclusion criteria but does not explain how the final sample sizes ($n = 20$ main; $n = 18$ pRF) were determined. Indicate whether these numbers were based on power analysis or prior work. Because valid runs per participant varied widely, explain how this variability was accommodated (e.g., weighting, run thresholds). Finally, clarify why 13 participants completed two sessions while eight completed one—was this a design decision, dropout, or exclusion? Note any fMRI specific exclusions (e.g., excessive motion) that shaped the final sample.**

Response: We appreciate the reviewer’s thoughtful observation regarding participant recruitment and session variability. The present study was conducted as part of a large-scale neuroimaging data collection project that involved concurrent acquisition of fMRI and camera-based eye tracking data. During the course of this project, we observed consistent inaccuracies in the eye tracking data, which led us to develop MoCET.

The participant sample reported in the original manuscript reflects the subset of participants available at the time of the initial submission ($N = 21$). One participant was excluded based on eye tracking quality criteria.

The original data collection project was structured to include both regular and extra sessions. **Participants were given the option to participate in the extra session after completing the regular session.** As a result, at the time of the original submission, 13 participants had completed both sessions (12 runs in total), while the remaining eight completed only the regular session (6 runs), either because they chose not to continue or because the extra session had not yet commenced.

To ensure consistency and robustness, **we restricted analyses to participants who completed all 12 runs.** Additionally, since the original submission, **five participants who had previously completed only the regular session also completed the extra session as part of our ongoing dataset collection project.** As a result, the updated dataset now includes 19 participants in total. After applying the same quality control criteria, 18 participants remained and were included in all analyses reported in the revised manuscript. All 18 participants also completed the pRF experiment session. As a result of this update, **approximately 33% more data now meet this criterion compared to the previous manuscript.** We also note that data exclusions were not made on the basis of fMRI data quality (see also Reviewer #3's comment 2).

We also conducted a power analysis based on the effect size observed in the original manuscript comparing MoCET and polynomial detrending methods. This analysis indicated that a sample size of at least 9 participants would be sufficient to achieve 80% power with Cohen's d of 1.14 at significance level of 0.05. Therefore, our revised sample size exceeds this requirement and ensures adequate statistical sensitivity.

“Nineteen participants (8 females, mean age 24.02 ± 2.54 years) were recruited for the study. All participants provided written informed consent, with the study protocol approved by the Institutional Review Board of Sungkyunkwan University, and received monetary compensation. Participants completed 12 runs of the Minecraft task across two or three fMRI sessions, depending on individual availability. Eye tracking data were evaluated based on specific validation criteria (see Methods—Eye tracking accuracy). One participant was excluded from the analysis as none of their runs met these criteria. As a result, data from 18 participants (average number of verifiable runs: 7.39 ± 3.03 , see Fig. 1c) were included in the final analysis. All 18 participants also completed an additional fMRI session involving pRF experiments, which served as a reference for validating the retinotopic mapping results.” —Methods (page 22)

“We implemented a two-step quality control procedure to distinguish different sources of eye tracking inaccuracy, such as instrumental error, fixation instability during calibration, and head motion-induced error. Specifically, we evaluated gaze accuracy within both the calibration and validation periods using separate models trained and tested on data from the same period. Runs with gaze error below 1.0° in both periods were labeled as verifiable and included in subsequent analyses related to eye tracking accuracy (e.g., Fig. 4a). If only the calibration phase met this threshold, the run was labeled as usable and retained for analyses that relied only on calibration (e.g., behavioral prediction and visual field mapping; Figs. 4b and 4c). Runs that did not meet the threshold in either phase were labeled unusable but still contributed to across-run generalization analyses (e.g., Fig. 5c), where no within-run calibration was required. The proportions of verifiable, usable, and unusable runs are shown in Fig. 1c. No data were excluded on the basis of fMRI data quality.” —Methods (page 26)

6. To illustrate the generalisability of MoCET, I recommend two supplementary transfer-tests. First, assess "cross-participant" transferability by applying each participant's regression coefficients to all other participants' eye-tracking data, then report the resulting gaze error relative to (i) participant-specific MoCET, (ii) polynomial detrending, and (iii) the uncorrected signal; this will quantify how strongly the model depends on individual camera-to-eye geometry. Second, evaluate "within-participant", across-session stability by re-using coefficients estimated in session 1 to correct session 2 (acquired on a different day), and compare their performance with coefficients re-estimated from session 2's own calibration; if cross-day error increases only marginally, it would highlight MoCET's practical advantage of shortening or omitting calibration in follow-up scans.

Response: Regarding cross-session generalizability, we agree that a generalizable correction method should ideally support calibration transfer across sessions. However, in our current dataset, across-session correction is limited by practical constraints. Specifically, we did not enforce a strictly identical camera setup or imaging volume alignment across sessions. Consequently, head motion parameters may not reflect a consistent displacement of the eye relative to the camera, undermining correction accuracy. We now explicitly acknowledge this limitation in the Discussion section of the revised manuscript.

Instead, **we implemented and evaluated an across-run version of MoCET, which performs calibration transfer across multiple runs within the same session.** Since camera placement and the imaging field-of-view are stable within a session, this setup allows for accurate motion-based correction using a shared reference volume. Our results demonstrate that across-run MoCET maintains robust correction accuracy without requiring run-specific calibration, offering a practical advantage for fMRI studies where within-run calibration is unavailable or unreliable. Moreover, this approach substantially reduces **calibration demands for follow-up runs, effectively shortening experimental duration without sacrificing data quality.** This functionality has now been added to the MoCET package, and its performance is presented in the revised manuscript and Figure 5.

We also note that the reviewer's comment was instrumental in motivating the development of the across-run MoCET extension.

Regarding cross-participant transferability, we agree this is a valuable direction for future work. However, MoCET's core assumption relies on participant-specific camera-to-eye-to-head geometry, which varies across individuals. Applying one participant's regression model to another would likely introduce substantial error due to differing eye positions, facial anatomy, and camera angles. For this reason, we did not include cross-subject transfer in the current evaluation but will consider it in future studies where these variables are better controlled or explicitly modeled.

“To enhance flexibility, we introduce an across-run variant of MoCET that enables calibration transfer across runs within the same session, allowing robust correction even when within-run calibration is limited or unavailable.” —Introduction (page 4)

“To evaluate whether MoCET can also correct across-run motion drift, we trained a motion-to-gaze regression model on a single calibration run (the run with the highest-quality calibration

among the six) and applied it to all other runs using head motion parameters realigned to the first volume of the calibration run (Fig. 5b). As comparison baselines, we tested two alternatives: (1) a re-centering approach that re-aligns the calibration origin to the pupil center at the start of each new run, mimicking the manual drift correction used in eye tracking software; and (2) Re-centering and polynomial detrending of the raw gaze time series.

Our results show that across-run MoCET slightly underperforms within-run MoCET in overall accuracy, yet significantly outperforms both detrending and re-centering methods (Fig. 5c, all $p < 0.001$). While re-centering can partially address translational drift, it fails to account for within-run motion and rotational shifts, limiting its effectiveness. Notably, across-run MoCET achieved robust performance even when runs were categorized as “unusable,” suggesting that effective across-run correction is possible despite suboptimal or missing within-run eye tracking calibration (Fig. 5c). These findings highlight MoCET’s potential to reduce experimental burden by requiring only a single calibration run per session, while preserving reliable gaze estimation across multiple runs.” —Results (page 16)

“Beyond improving within-run eye tracking accuracy, MoCET also generalizes effectively across multiple fMRI runs using a single calibration. By leveraging head motion parameters aligned to a shared reference volume, MoCET enables accurate correction of inter-run drift without the need for recalibration, reducing experimental burden and facilitating application to datasets with limited or inconsistent calibration. Nevertheless, while MoCET mitigates the need for frequent recalibrations, we caution against relying solely on a single calibration per session. If the only calibration run fails due to participant movement or tracking instability, recovery may be difficult. Thus, collecting multiple calibration runs throughout the session remains a recommended best practice to safeguard data quality and ensure fallback options when needed.

Moreover, MoCET can be extended to enable drift correction across sessions, offering the potential for longitudinal eye tracking alignment without session-specific calibration. However, this application requires strict control of experimental conditions, including consistent scanner setup, identical camera configuration, and alignment of imaging coverage and reference volumes across sessions. Without such consistency, head motion parameters may not accurately reflect the relative displacement between the eye and camera, undermining correction accuracy.” —Discussion (page 19)

“To test whether MoCET can correct inter-run eye tracking drift without repeated calibrations, we implemented an across-run correction protocol. For each participant, a single calibration run was selected (the run with the lowest validation error), and a linear regression model was trained to map 6-DoF head motion parameters to pupil displacement. To extend this model across runs, head motion parameters in the other runs were realigned to the same reference volume (i.e., the first volume of the calibration run), enabling prediction of motion-induced drift across separate scan runs. The learned model weights from the calibration run were then directly applied to all other runs without re-fitting.

In our full model, MoCET combines head motion regressors with low-order polynomial terms to capture both motion-induced and non-motion-related error. Since polynomial terms are time-dependent and cannot be reused across runs, we applied across-run MoCET in three steps: (1) apply motion correction using trained weights from the calibration run, and (2) fit new polynomial detrending per run to remove residual drift. We compared across-run MoCET to two baselines: (1) re-centering the gaze origin at the start of each run, mimicking manual drift

correction, and (2) re-centering followed by polynomial detrending to reduce slow drift. Performance was assessed using validation error across non-calibration runs. We also evaluated performance separately for verifiable and unusable runs (based on calibration quality) to assess robustness of MoCET to poor or missing calibration data.” —Methods (page 28)

—Figure 5 (page 15)

7. The terms “drift,” “error,” and “noise” are used interchangeably throughout the manuscript. Please define them consistently at the outset and use the preferred terms thereafter.

Response: We thank the reviewer for pointing out this issue. In the revised manuscript, we have standardized our terminology for clarity and consistency. Specifically, we now use “drift” exclusively to describe systematic changes in head position over time that contribute to camera-eye misalignment during fMRI experiments. We use “error” to refer to gaze inaccuracies. We have removed the term “noise” to avoid ambiguity. These definitions have been clarified early in the Methods section, and the revised terminology is now used consistently throughout.

8. Given the limited review window, I did not have capacity to inspect the submitted code line-by-line; instead, I concentrated on the conceptual soundness of the design, statistics, and interpretation. Because my comments request new analyses (e.g., a non-linear correction model and a sampling-rate sensitivity test), the code will necessarily change in the revision, at which point a full code audit would be more appropriate.

Response: We appreciate the reviewer’s thoughtful evaluation. In response to the reviewer’s comments and suggestions, we have substantially updated the MoCET Python package to include:

1. Nonlinear correction models, including variants with squared terms, temporal derivatives, and pairwise interaction terms.

2. Across-run MoCET, which supports calibration transfer across multiple fMRI runs within the same session.
3. Expanded data compatibility, including support for EDF files from the EyeLink system and motion parameters derived from FSL-based preprocessing pipelines.

All updated code and documentation are available in the latest version of the repository. We thank the reviewer for encouraging these improvements, which we believe substantially enhance MoCET's robustness and flexibility.

Reviewer #2

1. **I co-reviewed this manuscript with one of the reviewers who provided the listed reports. This is part of the Nature Communications initiative to facilitate training in peer review and to provide appropriate recognition for Early Career Researchers who co-review manuscripts.**

Response: We thank constructive feedback provided, which has significantly helped us improve the clarity, rigor, and impact of our manuscript.

Reviewer #3

1. **I am curious how the drift in eye tracker error between calibration and validation, and the reduction in error from MoCET, compare to standard within-run drift you see in standard eye tracking setups outside MRI scanners with head rests and/or bite bars. Relatedly, I'm curious for the behavioral prediction analysis what the prediction performance would be for standard eye tracking setups. This could be addressed by repeating the in-scanner experiments on a standard eye tracking setup. I'm not familiar with the brand of eye tracker used for the in-scanner experiments, but I don't think the out-of-scanner tracker would necessarily need to exactly match the in-scanner tracker. Ideally it's a setup with a chin rest for head stabilization, as this in my experience is the most common setup for out-of-scanner head stabilization in eye tracking. The benefits of this experiment are twofold.**
 - a. **First, it provides a reference frame for the in-scanner results. In the section "Impact of head motion on gaze tracking accuracy" and Figure 1, showing the calibration and validation error for an out-of-scanner eye tracker would provide a useful anchor for the scale of the problem for in-scanner eye tracking by showing the magnitude of the problem for in-scanner setups vs out-of-scanner setups. In the section "Compensating for head movement with Motion-Corrected Eye Tracking" and Figure 4a, the validation error for a standard out-of-scanner eye tracker would provide a floor for the error that indicates the degree of success. If MoCET reduces the validation error such that it is statistically the same as the validation error for an out-of-scanner eye tracking**

setup, that provides strong evidence that MoCET has functionally solved the targeted problem.

- b. Second, it provides a reference frame for the behavioral prediction performance in the section “MoCET enables high-precision eye tracking during free-viewing experiments” and Figure 4B. As currently presented, it’s clear that MoCET allows some prediction performance improvements over standard in-scanner eye tracking correction approaches, but I’m not sure whether the 15% hit rate is in line with what should be expected from standard head stabilized eye tracking.

Response: We thank the reviewer for this suggestion. In response, we conducted an additional behavioral eye tracking experiment outside the scanner using a stabilized laboratory setup (EyeLink 1000 Plus at 1000 Hz with a chin rest and head post) to benchmark MoCET’s performance against standard out-of-scanner conditions.

To establish a reference frame, we compared calibration and validation errors between in-scanner eye tracking data from fMRI experiments and behavioral eye tracking data collected outside the scanner. The results showed that validation error from motion-corrected, in-scanner eye tracking data was statistically comparable to that for out-of-scanner behavioral eye tracking, indicating that MoCET reduces drift to levels comparable to high-quality out-of-scanner setups. These results are reported in the revised manuscript (Fig. 4a).

We also evaluated behavioral prediction performance (hit rate) under the same task using the behavioral eye tracking data. The hit rate obtained from the behavioral eye tracking data was statistically comparable to that from in-scanner eye tracking data corrected by MoCET, supporting the conclusion that MoCET enables behavioral inference accuracy comparable to standard setups. This comparison is included in the revised manuscript (Fig. 4b).

We thank the reviewer again for this valuable suggestion, which has led to a substantial enhancement to the manuscript.

“We further validate MoCET by showing that its corrected gaze accuracy closely matches that of behavioral eye tracking data collected under standard laboratory conditions.” —Introduction (page 4)

“Notably, gaze accuracy achieved by MoCET was not statistically different from that observed in behavioral eye tracking experiments conducted outside the scanner, indicating that MoCET recovers high-quality gaze signals even under the constraints of fMRI environments ($t(20) = 1.34, p = 0.195$)” —Results (page 13)

“We also demonstrated that the hit rate from motion-corrected eye tracking was comparable to that observed in behavioral eye tracking, indicating reliable recovery of task-relevant gaze patterns during fMRI experiments ($t(20) = 1.40, p = 0.175$).” —Results (page 13)

“To further evaluate MoCET’s generalizability and accuracy, we analyzed three independent datasets: two movie-watching fMRI datasets collected at 3T and 7T scanners from separate groups of 12 participants each, and a behavioral eye tracking dataset from four participants who completed three runs of the Minecraft task outside the MRI scanner. The behavioral dataset

enabled a direct comparison between standard laboratory eye tracking and MRI-compatible eye tracking corrected using MoCET.” —Methods (page 23)

“For behavioral experiments outside the scanner, pupil position was recorded at 1000 Hz using the EyeLink 1000 Plus (SR Research) and subsequently downsampled to 100 Hz for analysis” —Methods (page 23)

—Figure 4 (page 12)

- Page 23, “Eye tracking accuracy”, Paragraph 2 says that all data w/ calibration/validation errors of >1 degree were excluded from subsequent analysis “to exclude errors stemming from fixation instability or individual gaze variability, thereby ensuring that any remaining inaccuracies were not simply due to inconsistent eye movements.” I’m not sure how this could be the case, given that much larger validation errors are reported in the paper. In the first Results section and Figure 1, the reported validation error is 7.093 degrees and in Figure 4 the uncorrected validation errors are in the range of 4-7 degrees. Please clarify if I misunderstood and adjust the text so the exclusion criteria are clear and consistent with the main text.

Response: We thank the reviewer for this observation. Eye tracking inaccuracies can arise from several distinct sources, including instrumental error, fixation instability during calibration, and motion-induced drift. To accurately evaluate MoCET’s performance and distinguish these sources, we implemented a two-step quality control procedure.

Specifically, **we applied the calibration procedure separately to the calibration and validation periods, and computed within-period gaze error using the same data segment used for model training.** If both within-period errors were sufficiently low (i.e., below 1.0°), the data were deemed verifiable and used to compute the cross-period validation error, as this indicated stable participant fixation and the absence of substantial instrumental error.

Therefore, the large validation errors reported in Figure 1 are not in conflict with our exclusion criterion. **These values reflect the cross-period validation error** (i.e., after training the model on the calibration period and testing on the validation period). The exclusion criterion based on within-period accuracy ensures that such large validation errors are not simply attributable to inconsistent eye movements.

To further clarify this point, we have revised the manuscript to explicitly distinguish between within-period accuracy (used for classification and exclusion) and cross-period validation error (used for analysis). We have also updated Figure 1c to visualize this classification.

“We implemented a two-step quality control procedure to distinguish different sources of eye tracking inaccuracy, such as instrumental error, fixation instability during calibration, and head motion-induced error. Specifically, we evaluated gaze accuracy within both the calibration and validation periods using separate models trained and tested on data from the same period. Runs with gaze error below 1.0° in both periods were labeled as verifiable and included in subsequent analyses related to eye tracking accuracy (e.g., Fig. 4a). If only the calibration phase met this threshold, the run was labeled as usable and retained for analyses that relied only on calibration (e.g., behavioral prediction and visual field mapping; Figs. 4b and 4c). Runs that did not meet the threshold in either phase were labeled unusable but still contributed to across-run generalization analyses (e.g., Fig. 5c), where no within-run calibration was required. The proportions of verifiable, usable, and unusable runs are shown in Fig. 1c. No data were excluded on the basis of fMRI data quality.” —Methods (page 26)

—Figure 1 (page 5)

3. How big are the stimuli in the fMRI experiments in degrees of visual angle? I couldn't find this anywhere in the main text or the methods, and this is helpful for determining the size of the eye tracker errors (which are already reported in degrees) relative to the stimuli.

Response: We thank the reviewer for pointing this out and apologize for the oversight. While the visual angle of the stimuli used in the fMRI experiments was illustrated in Figure 2A, we agree that it should also be stated in the Methods. To address this, **we have revised the Methods to explicitly report stimulus size in degrees of visual angle.** Additionally, we have included visual angle information for the video-watching dataset acquired on the 3T MRI scanner (as suggested by Reviewer #4), as well as for the behavioral eye tracking dataset. We believe these additions provide clearer context for interpreting the magnitude of eye tracking errors relative to stimulus dimensions.

“For the main analysis, we used eye tracking data collected while participants performed a 13-min 3D Minecraft-based video game task in a 7T MRI scanner. The stimulus was presented on a screen subtending $20.5^\circ \times 12.8^\circ$.” —Methods (page 23)

“For comparison, a behavioral version of the Minecraft task was conducted outside the scanner, with the display size matched to the MRI setup in visual angle.” —Methods (page 24)

“To assess generalizability across tasks and MRI scanners, we analyzed two additional movie-watching fMRI datasets acquired from separate groups of 12 participants each, on 3T and 7T scanners. Each dataset included standard eye tracking calibration (24-point calibration at the beginning and 12-point validation at the end of each run). In the 3T dataset, participants viewed approximately 12 minutes of first-person-view movie clips on a screen subtending $27.0^\circ \times 15.2^\circ$. In the 7T dataset, participants watched two 10-minute runs of naturalistic movies on a screen subtending $20.5^\circ \times 12.8^\circ$.” —Methods (page 24)

4. **DeepMReye comparison. Does the fMRI data need to be normalized to MNI space for the DeepMReye model to run? Since the analyses are within-participant, normalizing to MNI is not essential for group-level comparison, and DeepMReye may show better performance in each participants native space.**

Response: We confirmed that the original implementation of DeepMReye was performed in MNI space, as explicitly stated in their reporting summary. Additionally, the pre-trained models and tutorials from DeepMReye use eyeball masks defined in MNI space, which are required to extract voxel signals from the eyeball region. For this reason, we spatially normalized the data to MNI space to ensure compatibility with their pipeline and to apply the published voxel selection procedure as intended. While DeepMReye may benefit from native-space decoding in future work, we followed the original implementation to enable a fair and reproducible comparison.

Reviewer 4

1. **Although the method is itself general and practical, the acquisition protocol perhaps less so. One would have liked to see generalization across more than one 7T dataset, one of which involved a fairly unusual task (Minecraft).**

Response: We agree and have addressed this concern by replicating our findings in two additional datasets: a 7T movie-watching dataset and a 3T video-watching dataset. These datasets use different tasks and scanner settings compared to the original 7T Minecraft dataset. In both datasets, MoCET consistently outperformed uncorrected data and conventional detrending methods, supporting its generalizability across tasks and acquisition protocols. These results are now described in the revised manuscript and shown in Supplementary Figure 4. We appreciate the reviewer for highlighting this point.

“Using high-quality eye tracking data collected during free-viewing tasks, we demonstrate that MoCET consistently outperforms both uncorrected data and conventional detrending methods across independent datasets involving different tasks and MRI scanners.” —Introduction (page 4)

“Consistent results were observed in two independent replication datasets (Supplementary Fig. 4).” —Figure 4 caption (page 12)

“To further evaluate MoCET’s generalizability and accuracy, we analyzed three independent datasets: two movie-watching fMRI datasets collected at 3T and 7T scanners from separate groups of 12 participants each, and a behavioral eye tracking dataset from four participants who completed three runs of the Minecraft task outside the MRI scanner. The behavioral dataset enabled a direct comparison between standard laboratory eye tracking and MRI-compatible eye tracking corrected using MoCET.” —Methods (page 23)

“To assess generalizability across tasks and MRI scanners, we analyzed two additional movie-watching fMRI datasets acquired from separate groups of 12 participants each, on 3T and 7T scanners. Each dataset included standard eye tracking calibration (24-point calibration at the beginning and 12-point validation at the end of each run). In the 3T dataset, participants viewed approximately 12 minutes of first-person-view movie clips on a screen subtending $27.0^\circ \times 15.2^\circ$. In the 7T dataset, participants watched two 10-minute runs of naturalistic movies on a screen subtending $20.5^\circ \times 12.8^\circ$.” —Methods (page 24)

—Supplementary Figure 4 (page 45)

- Head motion can be quite variable across subjects. There may be a relationship between the characteristics of head-motion across subject (more or less high-frequency events, variation is predominant motion type, etc.) that could affect the effectiveness of MOCET. If head-motion is more "drift-like" its will be more similar to the polynomial regressor approach. But if motion is more spiky, then these methods will diverge, an nonlinearities may be more important (or that robust regression might yield better results). In general more discussion of individual variability in head-motion would be desirable; especially in light of point 1 (generalization); for example, where different patient populations or age-groups might have different characteristic head-motion profiles.

Response: We agree with the reviewer that variability in head motion characteristics could influence correction performance. As noted in our response to Reviewer #1’s Comment 1, we examined the performance of both linear and nonlinear variants of MoCET across different motion profiles, including drift-like and spike-like motion. We found no significant

advantage of nonlinear models across head motion types. As the reviewer suggested, polynomial regressors performed better specifically under drift-like motion, highlighting that motion characteristics can influence correction performance. We acknowledge the importance of individual variability and have expanded discussion to address this point in the revised manuscript.

“To further test the linear assumption, we evaluated the nonlinearity in the relationship between head motion and pupil coordinates across different types of head motion. We found that the linear and nonlinear variants of MoCET performed comparably across motion categories. Additional simulations with artificially amplified motion indicated that the linear MoCET model remained effective at motion levels typical of fMRI studies, while nonlinear models offered only limited additional benefit under extreme motion (for details, see Supplementary Fig. 3 and Methods)” —Results (page 11)

“MoCET assumes a linear relationship between head motion and gaze error, which may become inadequate in cases of extreme head movement or when the eye tracking camera is installed at a steep tilt relative to the gaze direction. In such conditions, nonlinear distortions in pupil coordinates can introduce errors not fully accounted for by linear regression. However, our empirical evaluation across diverse head motion types, including large, spike-like, and drift-like movements, showed that the linear MoCET model performed as well as more complex nonlinear variants. Furthermore, simulations with systematically amplified motion confirmed that linear MoCET remains effective across a broad range of motion levels. Nonlinear extensions provided improvements only under extreme head motion conditions, which are rarely encountered in empirical experiments (Supplementary Fig. 3). These findings support the validity of the linear assumption in typical fMRI settings, where head motion is often constrained by stabilization devices such as foam padding, minimizing extreme deviations that could introduce nonlinear distortions. To accommodate potential edge cases, we also provide these nonlinear MoCET variants in our Python package, allowing users to apply more flexible correction models when needed.” —Discussion (page 21)

“While MoCET assumes a linear relationship between head motion and drift in eye tracking data, this assumption may not hold under all circumstances. In particular, when head motion is large or abrupt, a simple linear regression model may be insufficient. To evaluate when the linearity assumption in MoCET remains valid and whether more complex, nonlinear models provide added value, we categorized eye tracking runs based on their head motion profiles and compared multiple versions of MoCET under these conditions. ‘Large motion’ runs were defined as those with a maximum in-plane displacement exceeding a +1 z-score threshold. To identify drift-like motion, we computed autocorrelations of each of the six head motion parameters over lags of one to five TRs, then averaged across lags and parameters. Runs with a z-scored mean autocorrelation above +1 were classified as drift. In contrast, spike-like motion was characterized by abrupt, transient movements, defined as runs with a z-scored mean autocorrelation below -1.

To test whether nonlinear modeling improves performance under different head motion types, we introduced two extended models. MoCET-Large incorporates squared terms and temporal derivatives of the head motion parameters to capture larger or more dynamic shifts. MoCET-Interaction includes pairwise interaction terms between head motion axes to account for cross-

dimensional effects. These models were evaluated alongside the original linear approach to assess the benefit of increased model complexity.

To examine the validity of the linear assumption under more extreme conditions, we performed a simulation-based analysis. Because large head movements are difficult to elicit reliably during empirical fMRI sessions, we artificially amplified head motion by scaling the original motion parameters by 20% to 360%. This approach allowed us to systematically test MoCET and its nonlinear variants under a wide range of motion intensities, including cases beyond typical experimental variability.” —Methods (page 29)

3. **Related to above: there is an inherent discrepancy between the high sampling rate of eye-tracking (e.g., 60 Hz) and the low sampling rate of fMRI-derived motion estimates (~0.5-1 Hz). Although the authors argue convincingly that head motion is dominated by low-frequency components, fast, transient movements (e.g., respiration-related) may not be perfectly corrected and could partially alias into the residual noise.**

Response: We appreciate the reviewer’s comment. As noted in our response to Reviewer #1’s Comment 3, we conducted additional analyses to assess the impact of temporal resolution mismatch between motion parameters and eye tracking. Our results demonstrate that MoCET maintains robust correction accuracy even when applied to data downsampled to 1Hz, suggesting that MoCET is robust to temporal resolution mismatches and remains effective even when gaze signals are corrected at the sampling rate of fMRI motion estimates.

Even fast, transient head movements can be captured in fMRI if they cause net displacement between consecutive volumes. We agree, however, that rapid, oscillatory motion occurring entirely within a single TR (e.g., a brief move-and-return) may not result in a detectable displacement in the brain images. In such cases, motion parameters from fMRI may underestimate the movement, reducing correction accuracy. We have explicitly acknowledged this limitation in the revised manuscript.

“However, rare, fast head movements within a single TR, such as a quick shift-and- return, may not produce detectable displacement in the fMRI images. In such cases, fMRI-derived motion parameters may underestimate the movement, reducing correction accuracy.” —Discussion (page 21)

Sincerely,
Jiwoong Park (jiwoongpark@skku.edu)
Jae Young Jeon (bluejay94@skku.edu)
Royoung Kim (royoungkim@skku.edu)
Kendrick N. Kay (kay@umn.edu)
Won Mok Shim (wonmokshim@skku.edu)